# META-CONTINUAL LEARNING OF NEURAL FIELDS

**Seungyoon Woo, Junhyeog Yun, Gunhee Kim**
Seoul National University
seungyoon.woo@vision.snu.ac.kr, {junhyeog,gunhee}@snu.ac.kr

## ABSTRACT

Neural Fields (NF) have gained prominence as a versatile framework for complex data representation. This work unveils a new problem setting termed *Meta-Continual Learning of Neural Fields* (MCL-NF) and introduces a novel strategy that employs a modular architecture combined with optimization-based meta-learning. Focused on overcoming the limitations of existing methods for continual learning of neural fields, such as catastrophic forgetting and slow convergence, our strategy achieves high-quality reconstruction with significantly improved learning speed. We further introduce Fisher Information Maximization loss for neural radiance fields (FIM-NeRF), which maximizes information gains at the sample level to enhance learning generalization, with proved convergence guarantee and generalization bound. We perform extensive evaluations across image, audio, video reconstruction, and view synthesis tasks on six diverse datasets, demonstrating our method's superiority in reconstruction quality and speed over existing MCL and CL-NF approaches. Notably, our approach attains rapid adaptation of neural fields for city-scale NeRF rendering with reduced parameter requirement. Code is available at https://github.com/seungyoon-woo/MCL-NF.

## 1 INTRODUCTION

Neural fields have recently emerged as a versatile framework for representing complex data with neural networks (Park et al., 2019; Mildenhall et al., 2020; Yariv et al., 2021; Yu et al., 2022). In this framework, a data point (e.g., an image, a video, or a 3D scene) is regarded as a field, and a neural network is trained to map a coordinate to the corresponding field value (e.g., pixel value, signed distance, radiance). Following the success of neural fields, multiple attempts have been made to learn neural fields in continual learning (CL) settings (Yan et al., 2021; Guo et al., 2023; Wang et al., 2023; Chung et al., 2022). The primary assumption of these approaches, which we collectively refer to as continual learning of neural fields (CL-NF), is that the information of a field is not fully accessible at once but instead sequentially obtained. Naively training a neural network on such a non-stationary data stream would result in catastrophic forgetting (Robins, 1995; Kirkpatrick et al., 2016), and thus additional techniques are introduced to mitigate the forgetting.

Meanwhile, one of the significant challenges of neural fields is the extensive training time. Rapid training of neural fields is crucial for practical applications, and it has motivated the emergence of meta-learning approaches for neural fields (ML-NF) (Tancik et al., 2021; Chen & Wang, 2022; Kim et al., 2023). Unlike conventional neural field training, which fits a neural network to a single data point, the meta-learning approaches consist of multiple data points split into meta-training and meta-test sets. By optimizing the learning process using the meta-training set, they can significantly reduce the training time of neural fields for new tasks.

In this work, we explore a novel problem setting that aims to combine the best of both worlds: **M**eta-**C**ontinual **L**earning of **N**eural **F**ields (MCL-NF). Within this setting, we meta-learn how to learn neural fields not only continually but also rapidly. Although there are several meta-continual learning (MCL) approaches in image classification domains (Javed & White, 2019; Beaulieu et al., 2020; Banayeeanzade et al., 2021), to the best of our knowledge, ours is the first work that explores MCL in the context of neural fields.

MCL-NF is powerful in drone and satellite applications, where large-scale environments are captured by resource-constrained edge devices and the memory limitation hinder full exploitation of

captured data. MCL-NF can leverage neural fields' ability to compress complex 3D environments into compact representations and continual learning's capacity to seamlessly transition between different levels of detail based on viewpoint and resources; as a result, MCL-NF offers fast learning, scalability, real-time rendering, and adaptability. This powerful combination facilitates fast generation of highly detailed virtual models for applications like urban planning, construction, environmental monitoring, and disaster response. We experiment our MCL-NF on real-world scenes in city-scale such as MatrixCity (Li et al., 2023) to shed a potential light on large-scale renderings.

As a first solution to MCL-NF so far, we explore a strategy that combines modular architecture with optimization-based meta-learning. Existing CL-NF approaches mostly rely on replaying previous data to prevent forgetting. However, replay-based approaches suffer from consistently increasing replay costs, which can be problematic in the MCL-NF setting where rapid adaptation to large-scale rendering is desired. Modularization is a major branch of CL that circumvents this problem by assigning a separate module for each task (Rusu et al., 2016; Aljundi et al., 2017). Moreover, modular architectures have been popularly studied in the neural field literature, especially for NeRF (Mildenhall et al., 2020), to improve the efficiency of handling large-scale scenes (Reiser et al., 2021; Turki et al., 2022; Mi & Xu, 2023). By learning a good initialization for the modules with an optimization-based meta-learning such as MAML (Finn et al., 2017), our approach can rapidly learn new tasks while maintaining existing knowledge. Previously, modularization techniques employed a fixed number of sub-modules for processing, despite the potential advantages of a continual learning that allows separating modules in a task-wise manner. By sharing initializations between sub-modules, we can increase the number of sub-modules whenever stakeholders require. This continual approach to modularization can dynamically adapt to evolving requirements without being constrained by a predetermined, finite number of modules.

We also introduce a Fisher Information Maximization loss for Neural Radiance Fields (FIM-NeRF) as a novel approach that incorporates Fisher Information directly into the NeRF training process. FIM-NeRF weights individual samples in the loss function based on their Fisher Information contribution, effectively prioritizing more informative regions of the scene during training. FIM-NeRF maximizes the mutual information between the model parameters and the observed data, potentially leading to more efficient learning and better generalization. We also present the theorems for proving the convergence guarantee and generalization bound of this new loss function. This sample-level application of Fisher Information distinguishes FIM-NeRF from previous parameter-level approaches in CL (Chaudhry et al., 2018; Konishi et al., 2023), offering a new perspective on how to leverage information theory in 3D scene representation and rendering.

Our approach, synergizing modular architecture with meta-learning, leads to no performance degradation incurred by forgetting during test-time. Our framework is versatile across various NF modalities. We carry out extensive empirical evaluation across the image, audio, video reconstruction, and view synthesis tasks on six diverse datasets, including three 2D image datasets (CelebA (Liu et al., 2015), FFHQ (Karras et al., 2019), and ImageNette (Deng et al., 2009)), one video dataset (VoxCeleb2 (Chung et al., 2018)), one audio dataset (LibriSpeech (Panayotov et al., 2015)), and one NeRF dataset (MatrixCity (Li et al., 2023)). Our method surpasses both previous MCL and CL-NF approaches in reconstruction quality and speed. Ultimately, our unified approach efficiently addresses city-scale NeRF challenges on MatrixCity (Li et al., 2023) using a smaller parameter size than existing single neural field methods.

## 2 RELATED WORKS

**Continual Learning of Neural Fields (CL-NF).** Neural fields learn a function $f_\theta : \mathbb{R}^d \to \mathbb{R}^q$ that maps a coordinate $x \in \mathbb{R}^d$ to a field quantity $y \in \mathbb{R}^q$ using a neural network to represent a complex data point. Continual learning trains a model on a series of tasks, $\mathcal{T}_1, \mathcal{T}_2, \ldots, \mathcal{T}_n$, each associated with its own data. In video examples, each frame can be a task $\mathcal{T}_i$, and in NeRF contexts, each grid separated by its 3D coordinates is a distinct task. Every task $\mathcal{T}_i$ consists of samples $\{(x_i^1, y_i^1), (x_i^2, y_i^2), \ldots, (x_i^m, y_i^m)\}$, each of which is made up of the coordinates. A significant challenge here is catastrophic forgetting (Robins, 1995; Kirkpatrick et al., 2016); when the model, after training on a new task $\mathcal{T}_{i+1}$, loses the knowledge acquired from the prior tasks.

Among several branches to handle forgetting, replaying previous samples has been a popular approach due to their simplicity and effectiveness. For the CL of neural fields, CNM (Yan et al., 2021)

utilizes experience replay to reconstruct a 3D surface from streaming depth inputs. Later, Guo et al. (2023); Chung et al. (2022); Po et al. (2023) leverage knowledge distillation to generate pseudo-samples of past experience using a neural network as memory storage. However, replay methods have scalability issues and increase computation in large-scale environments. Our proposed method is far from this limitation while solving challenges that arise in the CL problems.

**Meta-Learning of Neural Fields (ML-NF).** In the field of NF, several works leverage meta-learning for rapid learning and generality to unseen data instances and data-type agnostic features. MetaSDF (Sitzmann et al., 2020) learns a weight initialization for fitting neural representations to signed distance fields. LearnedInit (Tancik et al., 2021) and GradNCP (Tack et al., 2023) exploit bilevel optimizations to encode meta-initializations prior to expanding their horizons to general NFs. Modulating sub-parts of the architecture such as TransINR (Chen & Wang, 2022) and Generalizable IPC (Kim et al., 2023) can increase the efficiency of NF training. While they show promising results, they have to yet satisfy some applications like drone or satellite imagery (Po et al., 2023), which require sequential data processing with limited memory storage and immediate updates. Our approach introduces optimization-based weight initializations in CL settings, and effectively meets such demands of NFs while promoting quicker convergence and enhanced generality.

**Meta-Continual Learning (MCL).** Optimization-based meta-learning (Finn et al., 2017; Antoniou et al., 2019; Nichol et al., 2018; Fallah et al., 2020; Flennerhag et al., 2020) tries to find a favorable starting point of model's parameters to enhance the SGD optimization. However, a straightforward use of meta-learning makes all parameters vulnerable to SGD updates, which can lead to excessive adaptability and forgetting. Online meta-learning (Javed & White, 2019) is introduced as an MCL strategy to tackle this; in its inner loop, only the top layers are updated while fixing the other layers, which are updated in turn in the outer loop. ANML (Beaulieu et al., 2020) seeks to balance stability and plasticity in meta-learning; rather than freezing the lower part of the encoder, it uses a neuromodulatory network that remains static during the inner loop. However, they inherit some challenges from meta-learning, like the need to compute second-order gradients through the entire inner loop, leading to high computational demands and potential gradient issues.

**Modularization in Mixture-of-Experts.** Modularization can be viewed as a special case of the Mixture-of-Experts (MoE) framework (Jacobs et al., 1991), which has seen significant advancements in recent years particularly in the realm of large language models. For continual learning, Progressive networks (Rusu et al., 2016) and ExpertGate (Aljundi et al., 2017) are typical examples of this one-task-per-expert modularization strategy. Interestingly, Chen et al. (2023); Shen et al. (2023); Pan et al. (2024) optimize the distribution of data among experts by maximizing the mutual information (MIM) between the input and the expert selection. For instance, Pan et al. (2024) propose an MIM-based expert routing to improve the performance and interpretability of language tasks. In the context of neural fields, this has led to novel NeRF techniques for scene decomposition and representation (Rebain et al., 2021; Tancik et al., 2022). They leverage information-theoretic principles to guide the division of 3D space among different modules, effectively creating a modular representation of complex scenes.

Extending the concept of information-guided modularization to the meta-continual learning, we use Fisher Information Maximization (FIM) into our loss function, drawing parallels to the MIM approaches in MoE, but in the sample-specific level. This allows us to prioritize informative samples during training, potentially leading to more efficient learning and better generalization in the CL-NF.

## 3 PROBLEM STATEMENT

### 3.1 CONTINUAL LEARNING OF NEURAL FIELDS (CL-NF)

In the continual learning of NF, the model $f_\theta$ updates its parameters $\theta$ based on the sequential task-specific data. They are a series of context sets $C_i$, each comprising coordinates $(x^j, y^j)$ in $R^t, R^s$ for $j = 1, \ldots, m$. A context set can be defined as a subset of data relevant to a particular task within a sequence of tasks to be learned. While learning from a context set $C_i$, the model cannot access the previous context set $C_{i-1}$, adhering to the constraint of CL. Each context set functions as a distinct task $\mathcal{T}_i$ such as each frame in a video domain (Wang et al., 2023) or each 3D grid in a NeRF problem (Chung et al., 2022; Po et al., 2023) without explicit geometric supervision. The model $f_{\theta_i}$ updates its parameters based on the sequence of context $C_i$, producing an output $f_{\theta_i}^*$ for

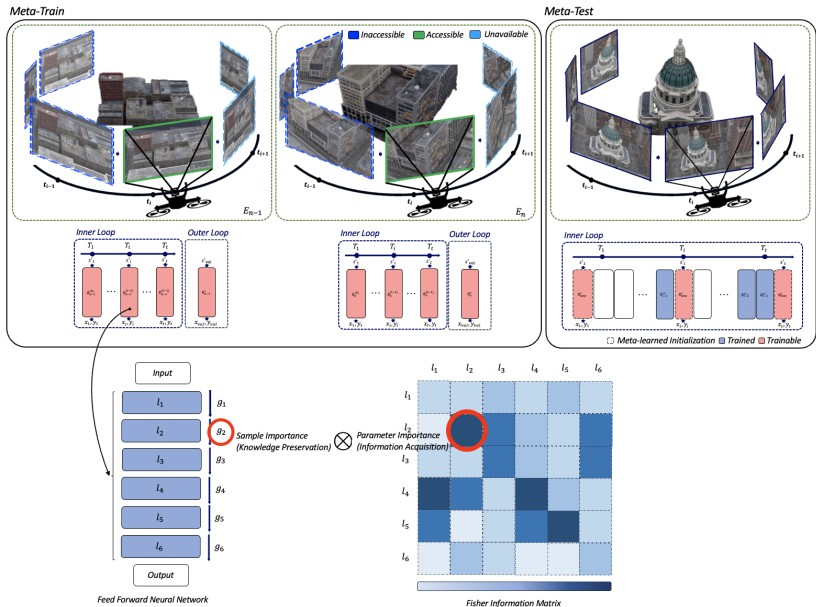

Figure 1: Illustration of the transition from traditional MSE loss to FIM-Loss in a neural network. It highlights how the FIM is used to calculate sample-specific weights. These weights are then incorporated into the final loss function, allowing the model to prioritize more informative samples.

$\mathcal{T}_i$, which then becomes the starting point for $\mathcal{T}_{i+1}$. The loss function for each $\mathcal{T}_i$ is defined as the Mean Squared Error (MSE): $\mathcal{L}_i(f_{\theta_i}) = \frac{1}{m} \sum_{j=1}^{m} (f_{\theta_i}(x_i^j, y_i^j) - s_i^j)^2$, where $f_{\theta_i}(x_i^j, y_i^j)$ predicts the signal value for coordinates $(x_i^j, y_i^j)$, and $s_i^j$ is the true value. The objective is to minimize the mean loss across all tasks up to $t$: $\mathcal{L}_{\text{total}} = \frac{1}{t} \sum_{i=1}^{t} \mathcal{L}_i(f_{\theta_i})$, which aims to reduce the cumulative loss and thereby mitigating catastrophic forgetting across all previously seen tasks.

## 3.2 MCL-NF: CL-NF MEETS META-LEARNING

Continual learning in the NF domain often requires a large number of convergence steps (Tack et al., 2023; Yan et al., 2021; Chung et al., 2022) and lacks generality across tasks (Guo et al., 2023; Wang et al., 2023). A prevalent solution may be to adopt optimization-based meta-learning, which not only addresses the challenges of convergence and generality but also brings forth the benefits of well-defined task variants and modality-agnostic features (Nichol et al., 2018; Zintgraf et al., 2019; Antoniou et al., 2019) .

We structure our approach into episodes $(\mathcal{E}_1, \mathcal{E}_2, \ldots, \mathcal{E}_n)$, each consisting of two stages: task-specific adaptation and meta-update. The task adaptation stage, akin to upon mentioned CL-NF, comprises sequences of signals forming context sets. For any given $\mathcal{T}_i$ in this stage, the model $f_{\theta_i}$ updates over $k$ iterations, starting from the end state of the previous $\mathcal{T}_{i-1}$, denoted as $f_{\theta_{i-1}^*}$, where $k$ typically spans hundreds of thousands of iterations in prior works (Yan et al., 2021; Chung et al., 2022). The meta-update stage optimizes the initial model configuration $f_{\theta_0}$ of each episode, based on the final output $f_{\theta_t^*}$. Here, the parameter $\theta'$ is optimized using

$$\theta' = \theta_0 - \eta \nabla_{\theta_0} \left( \frac{1}{t} \sum_{i=1}^{t} \frac{1}{m} \sum_{j=1}^{m} (f_{\theta_i}(x_i^j, y_i^j) - s_i^j)^2 \right), \tag{1}$$

where the initialization $\theta'$ is optimized by multiple meta-train episodes and becomes a starting point for the meta-test set, thereby enabling rapid adaptation to new tasks within meta-test episodes.

Meta-learning enables neural networks to assimilate priors for swift adaptation, thereby facilitating rapid learning and efficient memory usage. However, the mere conjunction of continual learning and meta-learning, like previous methodologies (Javed & White, 2019; Beaulieu et al., 2020), does not

---

**Algorithm 1** Modularized MCL-NF with Fisher Information Maximization Loss

---

**Require:** (1) A sequence of tasks $\{\mathcal{T}_1, \mathcal{T}_2, \ldots, \mathcal{T}_t\}$ along with context sets $\{C_1, C_2, \ldots, C_t\}$, (2) Learning rates $\eta_{\text{inner}}, \eta_{\text{outer}}$, and (3) Fisher Information weight $\lambda$.

1: **Shared Initialization:**
2: Initialize a shared set of parameters $\theta_{\text{shared}}$.
3: **for** $i = 1$ to $t$ **do**
4:      Initialize module $f_{\theta_i}$ with $\theta_{\text{shared}}$ for task $\mathcal{T}_i$.
5: **end for**
6: **for** $i = 1$ to $t$ **do**
7:      Select $m$ coordinates of a context set $C_i$ for task $\mathcal{T}_i$.
8:      Initialize Fisher Information Matrix $\mathbf{F}_i$ for task $\mathcal{T}_i$.
9:      **Inner Loop (Task-specific Learning):**
10:      **for** each training sample $(x, y)$ in context set $C_i$ **do**
11:          Compute output $\hat{s} = f_{\theta_i}(x, y)$.
12:          Compute Fisher Information weight: $w(\theta_i) = 1 + \lambda \cdot \text{tr}(\mathbf{g}(\theta_i)^T \mathbf{F}_i^{-1} \mathbf{g}(\theta_i))$,
13:             where $\mathbf{g}(\theta_i) = \nabla_{\theta_i} \log p(s|\hat{s})$.
14:          Compute FIM Loss: $\mathcal{L}_{\text{FIM}}(\theta_i; x, y) = w(\theta_i) \cdot \|\hat{s} - s\|^2$.
15:          Update $\theta_i$ for task $\mathcal{T}_i$:
$$\theta_i^{(k)} = \theta_i^{(0)} - \eta_{\text{inner}} \sum_{j=1}^{k} \partial \mathcal{L}_{\text{FIM}}(f_{\theta_i^{(j-1)}}(x, y), s) / \partial \theta_i^{(j-1)}.$$
16:          Update Fisher Information Matrix $\mathbf{F}_i$.
17:      **end for**
18:      **if** new task arrives **then**
19:          Expand module $f_{\theta_i}$ by adding new parameters $\theta_{\text{new}}$ initialized with $\theta_{\text{shared}}$.
20:      **end if**
21:      **Outer Loop (Meta-learning):**
22:      **for** each validation sample $(x_{val}, y_{val})$ in query set $Q_{val}$ **do**
23:          Compute FIM Loss $\mathcal{L}_{\text{FIM}}(\theta_i; x_{val}, y_{val})$ as in steps 10-12.
24:          Update $\theta_i$ for meta initialization: $\theta_i^* = \theta_i - \eta_{\text{outer}} \nabla_{\theta_i} \mathcal{L}_{\text{FIM}}(f_{\theta_i^{(k)}}(x_{val}, y_{val}), s_{val})$.
25:      **end for**
26: **end for**
27: **Inference:**
28: **for** each coordinate $(x, y)$ in the input space **do**
29:      Determine a subset of modules where $(x, y)$ belongs to: $G_i(x)$.
30:      Obtain the final output from relevant modules: $s(x, y) = \sum_{i=1}^{T} G_i(x) \cdot f_{\theta_i^*}(x, y)$.
31: **end for**

---

inherently ensure the mitigation of forgetting. Therefore, we propose a simple yet effective method to address these challenges.

# 4 APPROACH

## 4.1 SPATIAL AND TEMPORAL MODULARIZATION

We adopt MAML (Finn et al., 2017) as an optimization-based meta-learning algorithm, known for its efficiency with small convergence steps. Then, our method utilizes modularization within the MCL framework of NFs. The modular approach, inspired by the recent successful large-scale NeRFs such as MegaNeRF (Turki et al., 2022) and SwitchNeRF (Mi & Xu, 2023), not only ensures efficient memory usage with quicker computations but also mitigates catastrophic forgetting by keeping task-specific parameters separate. We can harness the intrinsic spatial and temporal correlations of coordinates, based on that subsequent frames in video or views in NeRF are highly overlapped. Thus, we enhance module adaptability by sharing initialization of each module, a key factor in large-scale radiance fields. Moreover, our shared initialization provides the flexibility to adjust the number of modules, independent of the number of tasks, which is vital for managing varying complexities within large datasets. Unlike the traditional MAML approaches that struggle with forgetting over extensive iterations in test-time, our strategy enables super-resolution image en-

hancements and detailed city-scale NeRF reconstructions while overcoming the spectral bias (Tack et al., 2023) often seen in NFs through effective test-time optimization.

## 4.2 THE FISHER INFORMATION MAXIMIZATION LOSS

For large-scale, resource-constrained continual learning, efficient knowledge re-use is crucial. However, standard modularization struggles with sharing knowledge across overlapping tasks such as repeating building appearances in city-scale 3D modeling. These limitations call for an additional method to complement modularization, facilitating efficient knowledge transfer, optimizing resources, and maintaining performance across tasks.

Hence, our approach takes advantage of sample re-weighting to replace popular rehearsal methods that may struggle when storage is limited and rapid processing is required. This approach dynamically adjusts the input importance based on information content, and prioritizes learning from the most relevant aspects of current inputs. Consequently, it can achieve similar knowledge retention and transfer as rehearsal methods, but without memory and computational overheads.

As modularization is viewed as a mixture of single-task experts, mutual information maximization can be a useful vehicle. Our solution is to perform sample re-weighting exploiting Fisher Information Matrix (FIM), which captures task-relevant information within model parameters without explicit storage of past experiences. It provides a computationally efficient proxy for mutual information through a local quadratic approximation of KL-divergence between parameter distributions.

FIM calculation from readily available gradient information facilitates efficient optimization, making it particularly suitable for our large-scale, resource-limited settings.

We propose the Fisher Information Maximization loss (FIM loss) to replace Eq.(1) by appending only weight term:

$$\theta' = \theta_0 - \eta \nabla_{\theta_0} \left( \frac{1}{t} \sum_{i=1}^{t} \frac{1}{m} \sum_{j=1}^{m} w_{ij}(\theta_i)(f_{\theta_i}(x_i^j, y_i^j) - s_i^j)^2 \right), \tag{2}$$

and $w_i(\theta)$ is a weight derived from the Fisher Information:

$$w(\theta_i) = 1 + \lambda \mathbf{g}(\theta_i)^T \mathbf{F}(\theta)^{-1} \mathbf{g}(\theta_i), \tag{3}$$

where $\mathbf{g}_i(\theta) = \nabla \theta \log p(s_i^j | f_\theta(x_i^j, y_i^j))$ is the score function, $\mathbf{F}(\theta)$ is the Fisher information matrix, and $\lambda > 0$ is a hyperparameter. The weight $w_i(\theta)$ comprises a base weight of 1, a hyperparameter $\lambda$ controlling Fisher Information influence, the gradient $\mathbf{g}_i(\theta)$ representing sample sensitivity, and the inverse Fisher Information Matrix $\mathbf{F}(\theta)^{-1}$ for parameter importance normalization. Eq.(3) prioritizes the samples causing big changes in important model parameters. Larger gradients indicate more informative samples, while $\mathbf{F}(\theta)^{-1}$ adjusts for varying parameter importance. This weighting focuses the samples most likely to improve model performance, balancing knowledge preservation with new information acquisition, which is crucial for meta-continual learning.

## 4.3 THE RELATION WITH INFORMATION GAIN MAXIMIZATION

We now establish the connection between our FIM loss and the principle of information gain maximization. Based on the theorem in (Kunstner et al., 2019) that links KL-divergence to Fisher Information, we can show that the FIM loss approximates mutual information maximization between model parameters $\theta$ and data $\mathcal{D}$. The theorem and its detailed proof is presented in Appendix A.1.

Let $p(\mathcal{D}|\theta)$ and $p(\mathcal{D}|\theta + \Delta\theta)$ be two probability distributions parameterized by $\theta$ and $\theta + \Delta\theta$, respectively. The Kullback-Leibler (KL) divergence between these distributions can be approximated as (Belghazi et al., 2018; Veyrat-Charvillon & Standaert, 2009):

$$KL[p(\mathcal{D}|\theta)\|p(\mathcal{D}|\theta + \Delta\theta)] \approx \frac{1}{2}\Delta\theta^T \mathbf{F}(\theta)\Delta\theta, \tag{4}$$

where $\mathbf{F}(\theta)$ is the Fisher Information Matrix. As $F(\theta)$ provides a local quadratic approximation of KL-divergence, by using $F(\theta)$ in our sample-level loss function, we implicitly measure and prioritize information content, optimizing mutual information without explicit calculation.

**Comparison with EWC**. EWC (Chaudhry et al., 2018) also use Fisher Information against catastrophic forgetting. It works at the parameter level, penalizing changes to important previous task parameters. On the other hand, our FIM loss operates at the sample level, dynamically weighting samples based on the Fisher Information contribution. This enables finer learning control, crucial for neural fields' spatial and temporal information locality. By prioritizing informative data regions or timepoints, it enhances rapid adaptation to new tasks in continual learning of complex 3D environments, while preserving previous knowledge.

### 4.3.1 CONVERGENCE ANALYSIS AND GENERALIZATION BOUND OF FIM-SGD

In Appendix A.2, we present the theorem and its derivation about the convergence of our Fisher Information Maximization via Stochastic Gradient Descent (FIM-SGD). This analysis guarantees the convergence of our FIM-SGD algorithm to the optimal parameters, with a rate that depends on the condition number of the problem and the variance of the stochastic gradients. This convergence property can further explain the effectiveness of our algorithm.

A generalization bound provides a probabilistic limit on the generalization error, which measures how well an algorithm performs on unseen data compared to training data (Pérez & Louis, 2020; Cao & Gu, 2020). We present the theorem and its derivation on a generalization bound for our FIM-loss in Appendix A.3. It shows that our FIM-Loss maintains good generalization properties with the bound scaling similarly to standard VC-dimension based bounds.

## 5 EXPERIMENTS

We evaluate the performance of our approach across image, video, audio, and NeRF datasets with high diversity and minimal redundancy. Moreover, we investigate the compute and memory usage of our approach in large-scale environments. Due to space limitation, we include additional details of experiments in Appendix, which covers ablation study, visualizations, and computation analysis. They provide a more comprehensive view of our method's performance and characteristics.

### 5.1 EXPERIMENTAL SETUP

We set the number of continual tasks to four, following prior works (Mi & Xu, 2023; Cho et al., 2022). This translates to four frames in the video domain and four 3D grids in the NeRF domain. We measure increased performance over multiple iterations from meta-initialization, focusing on the order of computations (steps). We run all experiments three times, reporting average performance until 500K outer steps, following (Tancik et al., 2021).

**Baselines**. LearnedInit (Tancik et al., 2021) is an ML-NF approach, which can be regarded as an upper-bound baseline (Meta (UB)) since it leverages batch processing while bypassing continual learning challenges. Other ML-NF methods such as (Chen & Wang, 2022; Kim et al., 2023) employ transformer-based hypernetworks, which are not applicable to our problem setting. We compare with OML (Javed & White, 2019) as a representative MCL baseline, and MAML (Finn et al., 2017) combined with CL setup, which is denoted by 'MAML+CL'. We select OML because it supports both classification and regression tasks, aligning with neural fields' regression output. We also include EWC (Kirkpatrick et al., 2016) and ER (Rolnick et al., 2019), which represent typical regularization-based and rehearsal-based CL approaches in our comparisons, as well as MER (Riemer et al., 2019), which exemplifies advanced CL baseline with bi-level optimization. Throughout the experiments, 'Ours (mod)' and 'Ours (MIM)' refer to the proposed MCL-NF method with modularization without or with MIM, respectively.

### 5.2 RESULTS OF IMAGE RECONSTRUCTION

We first experiment 2D image reconstruction using CelebA (Liu et al., 2015), FFHQ (Karras et al., 2019), and ImageNette (Deng et al., 2009) datasets. We process a 2D image by dividing it into multiple patches; for instance, a $180 \times 180$ image is split into four $180 \times 45$ patches. The coordinate-based MLP consists of five layers with Sine activations, configured with $d = 128$, $d_{in} = 2$ for image signal input, and $d_{out} = 3$ for RGB outputs. We measure the reconstruction accuracy for the entire images in terms of PSNR.

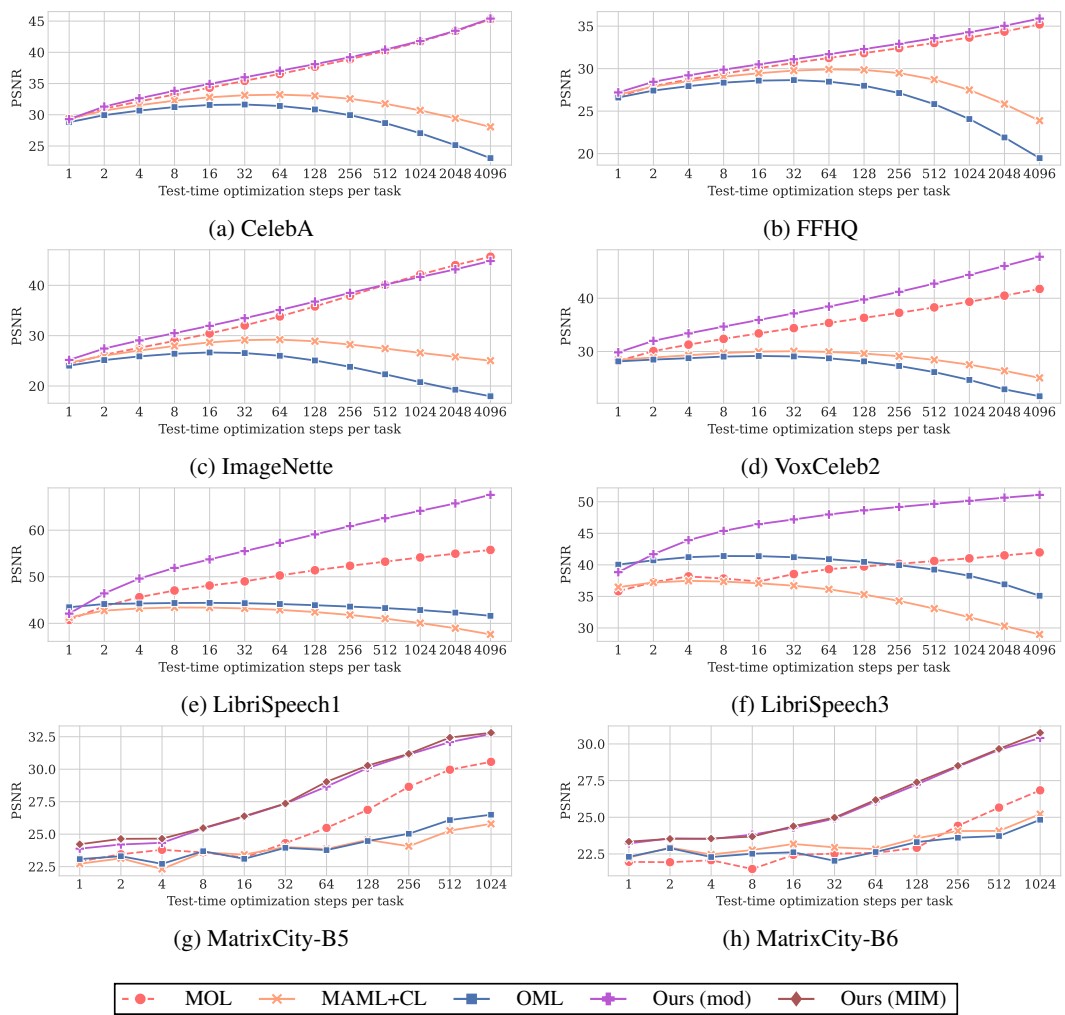

Figure 2: The PSNR comparison between various meta-learning methods over the adaptation steps. Our method demonstrates consistent improvement in PSNR as the number of steps increases, outperforming traditional MAML (Finn et al., 2017) and OML (Javed & White, 2019), particularly in longer adaptation sequences in all modalities and datasets.

**Highly-Structured Images.** The CelebA (Liu et al., 2015) has been actively used in meta-continual learning research. The consistent positioning of faces in the center of images provides a strong prior, ensuring a level of uniformity across the dataset. For CelebA, we use $178 \times 178$ resolution images with 1-pixel zero-padding for $180 \times 180$ input resolution, adhering to established practices.

**High-Resolution Images.** The FFHQ (Karras et al., 2019) consists of $512 \times 512$ high-resolution images. It evaluates whether even with limited memory meta-learning can reconstruct the data.

**High-Diversity Images.** The ImageNette (Deng et al., 2009) with a $160 \times 160$ resolution shows its high diversity, containing various classes, objects, and backgrounds. It can demonstrate the model's robustness to reconstruct images with less redundancy in patches in just a few learning steps.

**Results.** Fig.2a–2c show the results of three image benchmarks. The consistently superior performance demonstrates the robustness of our method, with each dataset highlighting unique benefits. The PSNR values of our method are stably high across all test-time optimization steps in CelebA and FFHQ, suggesting that our method can maintain its great quality despite computational constraints. Furthermore, results on ImageNette show gradual improvement, implying our method's capability to handle diversity and complexity effectively.

## 5.3 RESULTS OF VIDEO RECONSTRUCTION

We use the VoxCeleb2 dataset (Chung et al., 2018) for 3-dimensional video processing. Each video is dissected into consecutive frames with a size of $112 \times 112$, extracted at consistent time intervals to form a sequence. Following prior works, we utilize a five-layer MLP with Sine activations, configured with $d = 256$, $d_{in} = 3$ for $(x, y, t)$ coordinates, where $t$ represents the temporal dimension, and $d_{out} = 3$ for RGB outputs. Four sub-modules are divided along temporal axis, meaning that each sub-module configured with ( $d = 128$, $d_{in} = d_{out} = 3$) is allocated to each of the frame sequence. The goal of the test is to accurately reconstruct these sequential frames, thereby achieving a coherent and continuous representation of the video. The temporal aspect of videos is essential for models to understand not only static features but also their sequence evolving over time.

**Results.** As shown in Fig.2d, the consistent enhancement in PSNR at each step confirms that our method can capture and reconstruct the temporal dynamics of video sequences. This trend validates our method's efficacy in dealing with sequential data, where temporal correlations are crucial.

## 5.4 RESULTS OF AUDIO RECONSTRUCTION

Our framework is further extended to audio experiments with the LibriSpeech-clean dataset (Panayotov et al., 2015). We train our framework on randomly cropped segments of audio data. In previous works (Kim et al., 2023; Tack et al., 2023), the coordinate-based MLP is equipped with five layers using Sine activations ($d = 256$, $d_{in} = 1$ for the $(x, y)$ coordinates, and $d_{out} = 1$), amounting to the parameter size of ($p = 197, 120$). We transition from a singular MLP to a more modular architecture. We divided the model into four sub-modules, each having a reduced dimensionality of ($d = 128$) and cumulatively totaling ($p = 196, 704$) parameters. For evaluation, we reconstruct audio segments for accurate and coherent playback. Test samples are meticulously trimmed into segments of three seconds. By breaking down long audio sequences into smaller, manageable segments for continual learning, models can meta-learn the characteristics of speech more effectively.

**Results.** In Fig.2f, our method consistently increases its reconstruction quality whereas the OML loses its ability up to 64 steps and its baseline MAML suffers from the same issue within the CL setting. Moreover, our methods show comparable or even higher performance (in some time-steps) than LearnedInit, the upper bound (UB) of meta-learning. Our model can understand and generate speech by capturing the relations between phonemes and the subtleties within various speech attributes, leading to more natural and accurate speech processing.

## 5.5 RESULTS OF VIEW SYNTHESIS

We evaluate our approach to view synthesis in the context of large-scale city rendering in the Matrix-City dataset (Li et al., 2023). Each city scene is processed into a set of multi-view images captured from various viewpoints to ensure comprehensive coverage of the urban environment. The resolution of $1920 \times 1080$ pixels is resized into $480 \times 270$ due to its memory constraint. We partition the city blocks into meta-training and meta-test sets, allocating 70% of the total blocks to meta-training and the remaining 30% to meta-testing. Following standard NeRF practices, we employ a ten-layer MLP with ReLU activations, configured with $d = 1024$, $d_{in} = 5$ for $(x, y, z, \theta, \phi)$ coordinates, where $\theta$ and $\phi$ represent viewing angles, and $d_{out} = 4$ for RGB and density outputs. We reduce the original hidden dimension by a half, configured with $d = 512$, as the computation of second-order derivatives in longer sequences significantly increases memory requirements. Four sub-modules are divided along spatial axes, meaning that each sub-module configured with $d = 256$, $d_{in} = 5$, $d_{out} = 4$. Following MegaNeRF (Turki et al., 2022), we aim not for the SOTA performance but for the practicality and efficiency of our method in a city-scale volumetric rendering.

**Results.** Both Ours (mod) and Ours (MIM) show higher quality results compared to the MAML+CL. The MAML+CL approach exhibits a symptom of forgetting during inner adaptation, as reflected in the performance metrics. Throughout most of the experimental periods, Ours (MIM) attains better performance than Ours (mod). Additionally, Ours (MIM) displays a more stable increase in performance over time, a characteristic that is necessary for test-time optimization scenarios.

Table 1: Performance comparison in terms of PSNR (with the **best** and second best) at three key points: Step 1 (initial performance), Best (step), and Step 4096 (final performance). Methods are categorized into offline learning (**OL**), Continual learning (**CL**), continual meta-learning (**CML**), Meta-offline learning (**MOL**), and Meta-continual learning (**MCL**).

| Modality | | Image | | | | | | | | |
|---|---|---|---|---|---|---|---|---|---|---|
| Dataset | | CelebA | | | ImageNette | | | FFHQ | | |
| Metric (PSNR) | | Step 1 | Best (step) | Step 4096 | Step 1 | Best (step) | Step 4096 | Step 1 | Best (step) | Step 4096 |
| OL | OL | 11.78 | 38.15 (4096) | 38.15 | 12.16 | 38.75 (4096) | 38.75 | 12.78 | 32.98 (4096) | 32.98 |
| CL | CL | 12.41 | 15.60 (256) | 15.57 | 13.14 | 16.76 (64) | 16.49 | 13.74 | 16.35 (128) | 15.93 |
| | ER | 12.53 | 43.90 (4096) | 43.90 | 13.23 | **45.74 (4096)** | **45.74** | 14.87 | 35.30 (4096) | 35.30 |
| | EWC | 12.59 | 20.08 (256) | 18.19 | 13.23 | 19.77 (256) | 17.30 | 13.96 | 21.29 (512) | 19.25 |
| CML | MER | 11.90 | 21.97 (256) | 20.27 | 12.28 | 21.31 (256) | 19.28 | 12.88 | 23.09 (256) | 20.61 |
| MOL | MOL | 29.21 | 45.28 (4096) | 45.28 | 24.35 | 45.68 (4096) | 45.68 | 26.71 | 35.21 (4096) | 35.21 |
| MCL | MAML+CL | **29.38** | 33.22 (64) | 28.06 | 24.63 | 29.21 (64) | 25.03 | 26.94 | 29.91 (64) | 23.88 |
| | OML | 28.81 | 31.64 (32) | 23.06 | 24.04 | 26.66 (16) | 17.96 | 26.60 | 28.65 (32) | 22.23 |
| | **Ours** | 29.29 | **45.40 (4096)** | **45.40** | **25.18** | 44.83 (4096) | 44.83 | **27.20** | **35.89 (4096)** | **35.89** |

| Modality | | Audio | | | | | | Video | | |
|---|---|---|---|---|---|---|---|---|---|---|
| Dataset | | LibriSpeech1 | | | LibriSpeech3 | | | VoxCeleb2 | | |
| Metric (PSNR) | | Step 1 | Best (step) | Step 4096 | Step 1 | Best (step) | Step 4096 | Step 1 | Best (step) | Step 4096 |
| OL | OL | 29.63 | 48.44 (4096) | 48.44 | 29.63 | 38.61 (4096) | 38.61 | 12.93 | 36.10 (4096) | 36.10 |
| CL | CL | 31.07 | 32.83 (256) | 32.77 | 30.85 | 32.31 (32) | 31.34 | 15.00 | 19.80 (32) | 18.18 |
| | ER | 30.86 | 49.83 (4096) | 49.83 | 30.42 | 38.49 (4096) | 38.49 | 14.08 | 40.39 (4096) | 40.39 |
| | EWC | 30.98 | 32.92 (64) | 32.43 | 30.75 | 32.32 (64) | 30.91 | 15.13 | 20.82 (256) | 19.04 |
| CML | MER | 29.27 | 34.86 (2048) | 34.85 | 29.39 | 33.19 (1024) | 33.07 | 12.78 | 21.80 (256) | 20.49 |
| MOL | MOL | 40.74 | 55.76 (4096) | 55.76 | 35.83 | 41.97 (4096) | 41.97 | 28.21 | 41.75 (4096) | 41.75 |
| MCL | MAML+CL | 41.30 | 43.40 (8) | 37.63 | 36.47 | 37.47 (4) | 28.98 | 28.45 | 30.05 (32) | 25.04 |
| | OML | **43.46** | 44.40 (16) | 41.62 | **40.04** | 41.40 (8) | 35.11 | 28.14 | 29.16 (16) | 21.59 |
| | **Ours** | 42.09 | **67.58 (4096)** | **67.58** | 38.84 | **51.09 (4096)** | **51.09** | **29.84** | **47.80 (4096)** | **47.80** |

| Modality | | NeRF | | | | | |
|---|---|---|---|---|---|---|---|
| Dataset | | MatrixCity-B5 | | | MatrixCity-B6 | | |
| Metric (PSNR) | | Step 1 | Best (step) | Step 1024 | Step 1 | Best (step) | Step 1024 |
| CL | ER | 14.318 | 30.214 (3488) | 26.339 | 6.956 | 28.869 (3984) | 26.235 |
| | EWC | 17.241 | 28.811 (3904) | 24.781 | 10.442 | 28.457 (3856) | 26.25 |
| MOL | MOL | 22.901 | 30.569 (1024) | 30.569 | 21.961 | 26.837 (1024) | 26.837 |
| MCL | MAML+CL | 22.722 | 25.792 (1024) | 25.792 | 22.253 | 25.218 (1024) | 25.218 |
| | OML | 23.082 | 26.5 (1024) | 26.5 | 22.305 | 24.83 (1024) | 24.83 |
| | **Ours (mod)** | 23.885 | 32.712 (1024) | 32.712 | 23.217 | 30.407 (1024) | 30.407 |
| | **Ours (MIM)** | **24.223** | **32.804 (1024)** | **32.804** | **23.341** | **30.761 (1024)** | **30.761** |

## 6 CONCLUSION

We introduced MCL-NF, a framework combining modular architecture and optimization-based meta-learning for neural fields. Our approach, featuring FIM-Loss, effectively addresses catastrophic forgetting and slow convergence in continual learning. Experiments across diverse datasets show superior performance in reconstruction quality and learning speed, especially in resource-constrained environments. MCL-NF demonstrates potential for scalable, efficient neural field applications in complex, dynamic data streams.

**Limitations.** Despite its advantages, MCL-NF faces challenges: potential performance degradation over very long, dissimilar task sequences; increased memory usage with task number growth; and possible limitations in handling abrupt, significant data distribution changes. These areas present opportunities for future research to enhance the framework's real-world applicability.

## 7 ACKNOWLEDGEMENTS

This work was supported by Samsung Advanced Institute of Technology and the Institute of Information & Communications Technology Planning & Evaluation (IITP) grants funded by the Korea government (MSIT) (No. RS-2022-II220156, Fundamental research on continual meta-learning for quality enhancement of casual videos and their 3D metaverse transformation; No. RS-2019-II191082, SW StarLab; No. RS-2021-II211343, Artificial Intelligence Graduate School Program (Seoul National University)), and by the Institute of Information & Communications Technology Planning & Evaluation (IITP)–Information Technology Research Center (ITRC) grant funded by the Korea government (MSIT) (No. RS-2024-00437633). Gunhee Kim is the corresponding author.

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

# A  APPENDIX

## A.1  DETAILED PROOF OF FISHER INFORMATION MAXIMIZATION

**Theorem 1** (Fisher Information and KL-Divergence)**.** *Let $p(\mathcal{D}|\theta)$ and $p(\mathcal{D}|\theta+\Delta\theta)$ be two probability distributions parameterized by $\theta$ and $\theta + \Delta\theta$ respectively. The Kullback-Leibler (KL) divergence between these distributions can be approximated as:*

$$KL[p(\mathcal{D}|\theta)\|p(\mathcal{D}|\theta+\Delta\theta)] \approx \frac{1}{2}\Delta\theta^T\mathbf{F}(\theta)\Delta\theta \tag{5}$$

*where $\mathbf{F}(\theta)$ is the Fisher Information Matrix.*

*Proof.*  We begin with the definition of KL-divergence Zhu et al. (2016):

$$KL[p(\mathcal{D}|\theta)\|p(\mathcal{D}|\theta+\Delta\theta)] = \mathbb{E}_{\mathcal{D}\sim p(\mathcal{D}|\theta)}\left[\log\frac{p(\mathcal{D}|\theta)}{p(\mathcal{D}|\theta+\Delta\theta)}\right] \tag{6}$$

Expand $\log p(\mathcal{D}|\theta + \Delta\theta)$ using Taylor series around $\theta$ Jiang (2021):

$$\log p(\mathcal{D}|\theta+\Delta\theta) \approx \log p(\mathcal{D}|\theta) + \Delta\theta^T\nabla_\theta\log p(\mathcal{D}|\theta) + \frac{1}{2}\Delta\theta^T\mathbf{H}_\theta\Delta\theta \tag{7}$$

where $\mathbf{H}_\theta$ is the Hessian of $\log p(\mathcal{D}|\theta)$. Substituting this into the KL-divergence:

$$KL[p(\mathcal{D}|\theta)\|p(\mathcal{D}|\theta+\Delta\theta)] \approx -\mathbb{E}_{\mathcal{D}\sim p(\mathcal{D}|\theta)}\left[\Delta\theta^T\nabla_\theta\log p(\mathcal{D}|\theta) + \frac{1}{2}\Delta\theta^T\mathbf{H}_\theta\Delta\theta\right] \tag{8}$$

The first term vanishes as $\mathbb{E}[\nabla_\theta\log p(\mathcal{D}|\theta)] = 0$ Jiang (2021). The negative expectation of the Hessian is the Fisher Information Matrix:

$$\mathbf{F}(\theta) = -\mathbb{E}_{\mathcal{D}\sim p(\mathcal{D}|\theta)}[\mathbf{H}_\theta] \tag{9}$$

Therefore, we arrive at the desired approximation Barajas-Solano & Tartakovsky (2019); Spantini et al. (2015):

$$KL[p(\mathcal{D}|\theta)\|p(\mathcal{D}|\theta+\Delta\theta)] \approx \frac{1}{2}\Delta\theta^T\mathbf{F}(\theta)\Delta\theta \tag{10}$$

## A.2  CONVERGENCE ANALYSIS

We now analyze the convergence properties of our Fisher Information Maximization Stochastic Gradient Descent (FIM-SGD) algorithm.

**Theorem 2** (Convergence of FIM-SGD)**.** *Under the following assumptions:*

1. *The expected loss function $\mathcal{L}(\theta) = \mathbb{E}_{\mathcal{D}}[\mathcal{L}_{FIM}(\theta;\mathcal{D})]$ is $\mu$-strongly convex and has $L$-Lipschitz continuous gradients Gong & Ye (2014).*

2. *The stochastic gradients are unbiased estimates of the true gradient: $\mathbb{E}[\nabla\mathcal{L}_{FIM}(\theta;\mathcal{D})] = \nabla\mathcal{L}(\theta)$ Gong & Ye (2014).*

3. *The variance of the stochastic gradients is bounded: $\mathbb{E}[\|\nabla\mathcal{L}_{FIM}(\theta;\mathcal{D}) - \nabla\mathcal{L}(\theta)\|^2] \leq \sigma^2$ Gong & Ye (2014).*

4. *The Fisher Information Matrix $\mathbf{F}(\theta)$ is positive definite and its eigenvalues are bounded between $\lambda_{min}$ and $\lambda_{max}$ Wilson & Murphey (2014).*

*The FIM-SGD algorithm with a constant learning rate $\eta$ satisfying $0 < \eta < \frac{2\mu}{\lambda_{max} L^2}$ converges in expectation to the optimal parameter $\theta^*$:*

$$\mathbb{E}[\|\theta_t - \theta^*\|^2] \leq (1 - 2\eta\lambda_{min}\mu + \eta^2 L^2 \lambda_{max}^2)^t \|\theta_0 - \theta^*\|^2 + \frac{\eta\sigma^2\lambda_{max}^2}{2\lambda_{min}\mu - \eta L^2\lambda_{max}^2} \tag{11}$$

*where $t$ is the number of iterations.*

*Proof.* Let $V_t = \|\theta_t - \theta^*\|^2$ be our Lyapunov function. The update rule for FIM-SGD is:

$$\theta_{t+1} = \theta_t - \eta\mathbf{F}(\theta_t)^{-1}\nabla\mathcal{L}_{FIM}(\theta_t; \mathcal{D}_t) \tag{12}$$

where $\mathcal{D}_t$ is the mini-batch at iteration $t$. We analyze the expected decrease in $V_t$ Wilson & Murphey (2014):

$$\mathbb{E}[V_{t+1}|\theta_t] = \mathbb{E}[\|\theta_t - \eta\mathbf{F}(\theta_t)^{-1}\nabla\mathcal{L}_{FIM}(\theta_t; \mathcal{D}_t) - \theta^*\|^2|\theta_t] \tag{13}$$

$$= V_t - 2\eta(\theta_t - \theta^*)^T\mathbf{F}(\theta_t)^{-1}\nabla\mathcal{L}(\theta_t) + \eta^2\mathbb{E}[\|\mathbf{F}(\theta_t)^{-1}\nabla\mathcal{L}_{FIM}(\theta_t; \mathcal{D}_t)\|^2|\theta_t] \tag{14}$$

Using the $\mu$-strong convexity of $\mathcal{L}(\theta)$, we have:

$$(\theta_t - \theta^*)^T\nabla\mathcal{L}(\theta_t) \geq \mu\|\theta_t - \theta^*\|^2 \tag{15}$$

And from the bounded eigenvalues of $\mathbf{F}(\theta)$:

$$(\theta_t - \theta^*)^T\mathbf{F}(\theta_t)^{-1}\nabla\mathcal{L}(\theta_t) \geq \lambda_{min}\mu\|\theta_t - \theta^*\|^2 \tag{16}$$

For the last term, using the $L$-Lipschitz continuity of $\nabla\mathcal{L}(\theta)$ and the bounded variance of stochastic gradients Gong & Ye (2014):

$$\mathbb{E}[\|\mathbf{F}(\theta_t)^{-1}\nabla\mathcal{L}_{FIM}(\theta_t; \mathcal{D}_t)\|^2|\theta_t] \leq \lambda_{max}^2(L^2\|\theta_t - \theta^*\|^2 + \sigma^2) \tag{17}$$

Combining these inequalities:

$$\mathbb{E}[V_{t+1}|\theta_t] \leq (1 - 2\eta\lambda_{min}\mu + \eta^2 L^2\lambda_{max}^2)V_t + \eta^2\sigma^2\lambda_{max}^2 \tag{18}$$

Taking the total expectation and applying this recursively:

$$\mathbb{E}[V_t] \leq (1 - 2\eta\lambda_{min}\mu + \eta^2 L^2\lambda_{max}^2)^t V_0 + \frac{\eta\sigma^2\lambda_{max}^2}{2\lambda_{min}\mu - \eta L^2\lambda_{max}^2} \tag{19}$$

which completes the proof.

### A.3 GENERALIZATION BOUND

We now derive a generalization bound for our Fisher Information Maximization Loss.

**Theorem 3** (Generalization Bound for FIM Loss). *Let $\mathcal{H}$ be a hypothesis class with VC-dimension $d$, and $\mathcal{L}_{FIM}$ be our Fisher Information Maximization Loss. Then, with probability at least $1 - \delta$, for all $h \in \mathcal{H}$:*

$$|\mathcal{L}_{FIM}(h) - \hat{\mathcal{L}}_{FIM}(h)| \leq O\left(\sqrt{\frac{d\log(N/d) + \log(1/\delta)}{N}}\right) \tag{20}$$

*where $\mathcal{L}_{FIM}(h)$ is the true expected Fisher Information Maximization Loss and $\hat{\mathcal{L}}_{FIM}(h)$ is the empirical Fisher Information Maximization Loss.*

*Proof.* We begin by noting that our Fisher weights are bounded:

$$1 \leq w_i(\theta) \leq 1 + \lambda F_{max} \tag{21}$$

where $F_{max}$ is the maximum eigenvalue of $\mathbf{F}(\theta)$ over all $\theta$. Let $W = 1 + \lambda F_{max}$.

We can now apply the standard VC-dimension based generalization bound to our weighted loss function Yang & Honorio (2020); Luxburg & Schölkopf (2011). For any $\epsilon > 0$:

$$P \left( \sup_{h \in \mathcal{H}} |\mathcal{L}_{FIM}(h) - \hat{\mathcal{L}}_{FIM}(h)| > \epsilon \right) \leq 4\mathcal{S}(\mathcal{H}, 2N) \exp \left( -\frac{N\epsilon^2}{32W^2} \right) \tag{22}$$

where $\mathcal{S}(\mathcal{H}, 2N)$ is the shattering coefficient of $\mathcal{H}$ on a sample of size $2N$. Using the fact Luxburg & Schölkopf (2011) that $\mathcal{S}(\mathcal{H}, 2N) \leq (2N)^d$ for a hypothesis class with VC-dimension $d$, and setting the right-hand side to $\delta$, we get:

$$\epsilon = O \left( W \sqrt{\frac{d \log(N/d) + \log(1/\delta)}{N}} \right) \tag{23}$$

Noting that $W$ is a constant (with respect to $N$), we arrive at the stated bound Chen et al. (2021).

## B  QUANTITATIVE RESULTS

The results in 1 indicate that our Meta-Continual Learning method, specifically Ours (MIM), generally demonstrates better initial performance and faster convergence across all tested modalities. When compared to Offline Learning (OL) and Continual Learning (CL) approaches, our method shows a consistently strong start, suggesting effective initialization and continual adaptation. In terms of achieving the highest PSNR (Best Step), Ours (MIM) often performs at least as well as Meta-Offline Learning (MOL), with significantly fewer optimization steps. Furthermore, Ours (MIM) maintains competitive quality at later steps, indicating stability during the learning process. CL methods, on the other hand, require much more optimization steps to reach comparable performance levels, and even then, often fall short of Ours (MIM) in terms of reconstruction quality. This trend is consistent across image, audio, video, and NeRF datasets in resource-constrained environments where rapid training is essential. Overall, the findings suggest that Meta-Continual Learning can offer a robust alternative to more resource-intensive methods, without sacrificing reconstruction quality.

We implement SSIM (Structural Similarity Index Measure) (Wang et al., 2004) for the image and video domains, and PESQ (Perceptual Evaluation of Speech Quality) (Rix et al., 2001) for the audio domain. In the table, we focus on a meta-continual learning specifically designed for learning to continually learn neural radiance fields. We excluded the methods that are categorized as Continual Bi-level optimization (Gupta et al., 2020; von Oswald et al., 2021; Wu et al., 2024) based on (Son et al., 2023), since their settings differ from ours. MCL aims to create an initialization or strategy that quickly adapts to new tasks while retaining prior knowledge. In contrast, the latter focuses on adapting the meta-learner itself over time. Instead, we plan to experiment with multiple MAML variants (Li et al., 2017; Antoniou et al., 2019; Nichol et al., 2018). The SSIM values show that our approach (Ours) maintains strong visual similarity, achieving higher scores at Step 4096 across all image and video datasets. Similarly, PESQ for the audio domain reveals that our method provides consistent perceptual quality compared to others, especially for longer adaptation sequences (Step 4096). Notably, our approach outperforms baseline models in terms of both PSNR and perceptual quality metrics, highlighting its robustness in maintaining quality during sequential learning.

## C  ABLATION STUDY

Our ablation study compared four variants of our method: modular (mod) and Mutual Information Maximization (MIM) approaches, each with hidden dimensions of 256 and 512 4. Results show that

| Modality | | Image | | | | | | Video | | |
|---|---|---|---|---|---|---|---|---|---|---|
| Dataset | | CelebA | | | ImageNette | | | VoxCeleb2 | | |
| Metric (SSIM) ↑ | | Step 1 | Best (step) | Step 4096 | Step 1 | Best (step) | Step 4096 | Step 1 | Best (step) | Step 4096 |
| OL | OL | 0.329 | 0.969 (4096) | 0.969 | 0.248 | 0.98 (4096) | 0.98 | 0.347 | 0.967 (4096) | 0.967 |
| CL | CL | 0.313 | 0.519 (64) | 0.463 | 0.248 | 0.526 (128) | 0.477 | 0.365 | 0.611 (64) | 0.493 |
| | ER | 0.316 | 0.547 (64) | 0.504 | 0.273 | 0.555 (128) | 0.489 | 0.438 | 0.629 (64) | 0.505 |
| | EWC | 0.321 | 0.63 (256) | 0.503 | 0.251 | 0.617 (256) | 0.475 | 0.369 | 0.648 (256) | 0.526 |
| CML | MER | 0.332 | 0.711 (256) | 0.633 | 0.251 | 0.674 (256) | 0.581 | 0.348 | 0.68 (256) | 0.603 |
| MOL | MOL | 0.82 | **0.995 (4096)** | **0.995** | 0.729 | **0.997 (4096)** | **0.997** | 0.807 | 0.991 (4096) | 0.991 |
| MCL | MAML+CL | **0.822** | 0.92 (64) | 0.816 | 0.742 | 0.884 (32) | 0.789 | 0.815 | 0.888 (32) | 0.773 |
| | OML | 0.796 | 0.884 (32) | 0.65 | 0.696 | 0.81 (16) | 0.553 | 0.796 | 0.855 (16) | 0.599 |
| | Ours | 0.817 | **0.995 (4096)** | **0.995** | **0.751** | 0.996 (4096) | 0.996 | **0.857** | **0.996 (4096)** | **0.996** |

| Modality | | Audio | | |
|---|---|---|---|---|
| Dataset | | LibriSpeech1 | | |
| Metric (PESQ) ↑ | | Step 1 | Best (step) | Step 4096 |
| OL | OL | 1.03 | 1.82 (2048) | 1.82 |
| CL | CL | 1.02 | 1.12 (2048) | 1.12 |
| | ER | 1.02 | 1.13 (4096) | 1.13 |
| | EWC | 1.02 | 1.12 (2048) | 1.12 |
| CML | MER | 1.03 | 1.16 (2048) | 1.16 |
| MOL | MOL | 1.33 | 2.48 (4096) | 2.48 |
| MCL | MAML+CL | **1.37** | 1.62 (32) | 1.59 |
| | OML | 1.34 | 1.62 (512) | 1.59 |
| | Ours | 1.3 | **3.73 (4096)** | **3.73** |

Table 2: Evaluation results for various methods across multiple modalities (Image, Video, Audio) and datasets. Metrics such as SSIM (for Image and Video) and PESQ (for Audio) are reported for different steps of optimization. Best results are highlighted in bold.

MIM consistently outperforms the modular approach across all steps, regardless of dimension size, indicating its significant contribution to the method's effectiveness. Increasing the hidden dimension from 256 to 512 generally improves performance, especially in later steps, with the improvement more pronounced in the modular approach. Notably, the MIM variant with 256 dimensions often outperforms the modular variant with 512 dimensions, suggesting MIM achieves better results with fewer parameters. All variants demonstrate consistent improvement over steps, showcasing the method's continual learning capability. The performance gap between variants widens as the number of steps increases, highlighting the benefits of MIM and larger hidden dimensions in longer sequences. These findings underscore the importance of both the MIM component and appropriate dimensionality in our method's overall performance.

To demonstrate the scalability and effectiveness of the MIM method over longer sequences, we expanded the number of tasks to 5 and 10 5. The metrics were evaluated at steps doubling from 1 to 1024. Overall, "Ours (MIM)" consistently outperforms "Ours (mod)" in terms of PSNR, especially at later steps. This suggests that the Fisher Information Maximization (MIM) strategy contributes to improved reconstruction quality during extended optimization. This trend is observed in both the 5-task and 10-task scenarios.

# D RESOLUTION-REDUCED SEQUENTIAL OPTIMIZATION

In this experimental setup, we focus on dividing images into lower resolutions. Specifically, an $n \times n$ image is split into four smaller sub-images of resolution $(n//2) \times (n//2)$, where each sub-image maintains the same height and width as the original image but with half the resolution. These smaller sub-images are then optimized sequentially, enabling faster convergence by processing less complex data in each optimization step. Importantly, the total image area remains unchanged, and only the resolution is reduced while preserving essential content and structure for training.

| Modality | | Image | | | | | | Video | | |
|---|---|---|---|---|---|---|---|---|---|---|
| Dataset | | CelebA | | | ImageNette | | | VoxCeleb2 | | |
| Metric (PSNR) ↑ | | Step 1 | Best (step) | Step 4096 | Step 1 | Best (step) | Step 4096 | Step 1 | Best (step) | Step 4096 |
| OL | OL | 11.84 | 38.2 (4096) | 38.2 | 12.19 | 38.87 (4096) | 38.87 | 13 | _37.83 (4096)_ | _37.83_ |
| CL | CL | 17.63 | 26.3 (4096) | 26.3 | 16.7 | 26.67 (2048) | 26.67 | 17.53 | 33.77 (1024) | 33.39 |
| | ER | 17.63 | 26.5 (1024) | 26.47 | 16.7 | 26.69 (2048) | 26.66 | 17.53 | 33.79 (1024) | 33.44 |
| | EWC | 18.21 | 28.12 (4096) | 28.12 | 17.01 | 26.88 (1024) | 26.75 | 17.96 | 34.18 (4096) | 34.18 |
| CML | MER | 12.21 | 26.62 (4096) | 26.62 | 12.56 | 26.62 (2048) | 26.61 | 13.03 | 33.94 (4096) | 33.94 |
| MOL | MOL | 29.28 | 46.36 (4096) | 46.36 | 24.52 | **48.55 (4096)** | **48.55** | 28.9 | **44.21 (4096)** | **44.21** |
| MCL | MAML+CL | **33.12** | 34.12 (4) | 32.6 | **27.6** | 28.24 (2) | 26.83 | _33.65_ | 36.3 (16) | 34.09 |
| | OML | _32.55_ | 33.69 (4) | 33.14 | _26.96_ | 27.53 (2) | 26.85 | **34.05** | 36.67 (16) | 35.19 |
| | Ours | 24.22 | **48.94 (4096)** | **48.94** | 20.62 | _46.81 (4096)_ | _46.81_ | 24.81 | 42.31 (4096) | _42.31_ |
| Modality | | Image | | | | | | Video | | |
| Dataset | | CelebA | | | ImageNette | | | VoxCeleb2 | | |
| Metric (SSIM) ↑ | | Step 1 | Best (step) | Step 4096 | Step 1 | Best (step) | Step 4096 | Step 1 | Best (step) | Step 4096 |
| OL | OL | 0.329 | 0.969 (4096) | 0.969 | 0.248 | 0.98 (4096) | 0.98 | 0.347 | 0.967 (4096) | 0.967 |
| CL | CL | 0.469 | 0.915 (4096) | 0.915 | 0.344 | 0.892 (4096) | 0.892 | 0.438 | 0.93 (1024) | 0.918 |
| | ER | 0.469 | 0.916 (4096) | 0.916 | 0.344 | 0.893 (2048) | 0.892 | 0.438 | 0.93 (1024) | 0.919 |
| | EWC | 0.483 | 0.926 (4096) | 0.926 | 0.357 | 0.892 (2048) | 0.891 | 0.462 | 0.935 (2048) | 0.932 |
| CML | MER | 0.343 | 0.914 (4096) | 0.914 | 0.258 | 0.882 (4096) | 0.882 | 0.356 | 0.93 (4096) | 0.93 |
| MOL | MOL | 0.82 | _0.995 (4096)_ | _0.995_ | 0.729 | **0.997 (4096)** | **0.997** | _0.807_ | **0.991 (4096)** | **0.991** |
| MCL | MAML+CL | **0.912** | 0.933 (64) | 0.902 | **0.852** | 0.878 (32) | 0.826 | **0.913** | 0.95 (16) | 0.912 |
| | OML | _0.892_ | 0.925 (8) | 0.915 | _0.805_ | 0.836 (4) | 0.822 | **0.913** | 0.952 (16) | 0.925 |
| | Ours | 0.669 | **0.996 (4096)** | **0.996** | 0.532 | 0.996 (4096) | _0.996_ | 0.659 | _0.984 (4096)_ | _0.984_ |

Table 3: This table presents the results of the experiment on resolution-reduced sequential optimization for image, and video modalities. The table compares the performance of various methods, including "Ours," in terms of image and video reconstruction quality at different optimization steps (Step 1, Best Step, and Step 4096).

| Steps | 1 | 2 | 4 | 8 | 16 | 32 | 64 | 128 | 256 | 512 |
|---|---|---|---|---|---|---|---|---|---|---|
| Ours (mod; 256) | 23.885 | 24.205 | 24.334 | 25.436 | 26.332 | 27.342 | 28.663 | 30.089 | 31.124 | 32.088 |
| Ours (MIM; 256) | **24.223** | **24.64** | **24.655** | 25.471 | 26.387 | 27.362 | **29.031** | 30.281 | 31.186 | 32.437 |
| Ours (mod; 512) | 24.02 | 23.906 | 24.502 | 25.445 | 26.29 | **27.94** | **29.354** | 30.231 | 31.403 | 32.486 |
| Ours (MIM; 512) | **24.124** | **24.176** | **24.813** | **25.555** | **26.399** | 27.859 | 29.115 | **30.301** | **31.529** | **32.63** |

Table 4: Performance comparison of our methods across different model configurations (rows) and optimization steps (columns). In each row, 'mod' refers to modularization, and 'MIM' to modularization with mutual information maximization. The number next to 'mod' or 'MIM' indicates the hidden dimension and the ray batch size for each iteration.

The experimental results demonstrate that our resolution-reduced sequential optimization approach consistently outperforms other methods across image and video modalities. For the image datasets (CelebA, ImageNette), our method ("Ours") achieves the highest SSIM scores of 0.995 at step 4096, surpassing upper bound approaches such as MOL. Similarly, for video data (VoxCeleb2), "Ours" yields the best SSIM score of 0.996 at step 4096, again outperforming competitors. These results highlight the effectiveness of reducing image resolution while maintaining essential content, enabling faster convergence and improved reconstruction quality during extended optimization steps. The advantage of our method becomes more apparent in later steps, where it consistently delivers better results in terms of SSIM, confirming the benefits of the proposed optimization strategy.

| # of Tasks | Method | Step 1 | Step 2 | Step 4 | Step 8 | Step 16 | Step 32 | Step 64 | Step 128 | Step 256 | Step 512 | Step 1024 |
|---|---|---|---|---|---|---|---|---|---|---|---|---|
| 5 | **Ours (mod)** | *24.072* | *24.155* | ***24.484*** | *24.696* | *25.072* | ***25.631*** | *26.645* | *28.075* | *29.665* | *30.836* | *31.681* |
|   | **Ours (MIM)** | ***24.143*** | ***24.276*** | *24.394* | ***24.775*** | ***25.102*** | *25.61* | ***26.83*** | ***28.23*** | ***29.73*** | ***31.005*** | ***31.822*** |
| 10 | **Ours (mod)** | *23.415* | *23.514* | *23.618* | *24.001* | *24.298* | *24.768* | *26.037* | *27.62* | *28.88* | ***29.76*** | *31.069* |
|   | **Ours (MIM)** | ***23.493*** | ***23.548*** | ***23.783*** | ***24.119*** | ***24.464*** | ***24.793*** | ***26.238*** | ***27.667*** | *28.886* | *29.71* | ***31.098*** |

Table 5: Performance metrics (PSNR) for "Ours (mod)" and "Ours (MIM)" across different optimization steps, for 5 and 10 tasks. The metrics are evaluated at steps doubling from 1 to 1024.

# E    COMPUTATIONAL COST ANALYSIS

We note that the actual computational cost can vary significantly based on the specific hardware and the software implementation used. In this work, we evaluate the computational cost using our PyTorch implementation, conducted on NVIDIA TITAN X Pascal GPUs which have 12 GB of VRAM.

Table 6 presents a comparison of the number of parameters, GPU memory consumption (in MiB), test-time optimization speed (measured in episodes per second), and reconstruction quality (PSNR) on the CelebA dataset. An episode corresponds to a single image with a resolution of $180 \times 180$, and each episode consists of 4 tasks, with each task corresponding to an image patch of size $180 \times 45$. The reported test-time optimization costs are based on a batch size of 1, with 128 optimization steps per task, performed across a total of 4 tasks.

Our proposed method demonstrates a compelling balance of performance and efficiency in the meta-continual learning landscape. Despite a marginal increase in parameters (201.23K compared to 198.91K for other methods), our approach achieves the highest PSNR of 38.10, surpassing all other techniques including offline and meta-offline learning. Notably, it accomplishes this while utilizing the least GPU memory (787 MiB), significantly lower than the next best performer. The test-time optimization (TTO) speed of 68.27 ep/s is competitive, outpacing several other methods including traditional continual learning approaches. This combination of top-tier performance (PSNR) with minimal memory footprint showcases our method's efficiency in resource utilization. While not the fastest in terms of TTO speed, the substantial gains in output quality and memory efficiency make our approach particularly valuable for applications where high-quality results and resource constraints are critical factors, especially in continual learning scenarios.

| Category | Method | Parameters (↓) | GPU Memory (↓) | TTO Speed (↑) | PSNR (↑) |
|---|---|---|---|---|---|
| Offline Learning | OL | 198.91K | 1068 MiB | 65.51 ep/s | 25.18 |
| Continual Learning | CL | 198.91K | 989 MiB | 59.90 ep/s | 15.55 |
|  | ER | 198.91K | 991 MiB | 37.20 ep/s | 30.83 |
|  | EWC | 198.91K | 968 MiB | 70.45 ep/s | 19.70 |
| Continual Meta-Learning | MER | 198.91K | 890 MiB | 9.27 ep/s | 21.69 |
| Meta-Offline Learning | MOL | 198.91K | 1044 MiB | 79.02 ep/s | 37.69 |
| Meta-Continual Learning | MAML+CL | 198.91K | 977 MiB | 65.12 ep/s | 33.04 |
|  | OML | 198.91K | 1011 MiB | **160.29** ep/s | 30.85 |
|  | Ours | 201.23K | **787** MiB | 68.27 ep/s | **38.10** |

Table 6: Computational cost comparison.

| Method | Number of Networks | Hidden Dimension | Number of Layers | Parameter Count (with Bias) |
|---|---|---|---|---|
| Un-modularized | 1 | 512 | 10 | 2,628,099 |
| Modularized | 4 | 256 | 10 | $4 \times 659,459 = 2,637,836$ |

Table 7: Comparison of parameter counts for different network configurations, including un-modularized and modularized versions with bias terms included.

## F VISUALIZATIONS

Our proposed method demonstrates superior performance across diverse image datasets, consistently achieving the highest PSNR scores. In the CelebA dataset, our approach attains a PSNR of 45.4, significantly outperforming other methods, with the nearest competitor (ER) at 43.90. The visual quality of our result closely matches the ground truth, preserving fine facial details and overall image clarity. For ImageNette, our method again leads with a PSNR of 44.83, showcasing its ability to handle complex, non-facial images with high fidelity. In the challenging FFHQ dataset, which features high-resolution facial images, our method achieves a PSNR of 35.89, surpassing all other approaches. Notably, our results consistently show better preservation of intricate details, accurate color reproduction, and reduced artifacts compared to other methods. This performance is particularly impressive given that it maintains high quality across different image types and complexities, from facial features to intricate object details, demonstrating the robustness and versatility of our approach in continual learning scenarios for image reconstruction tasks.

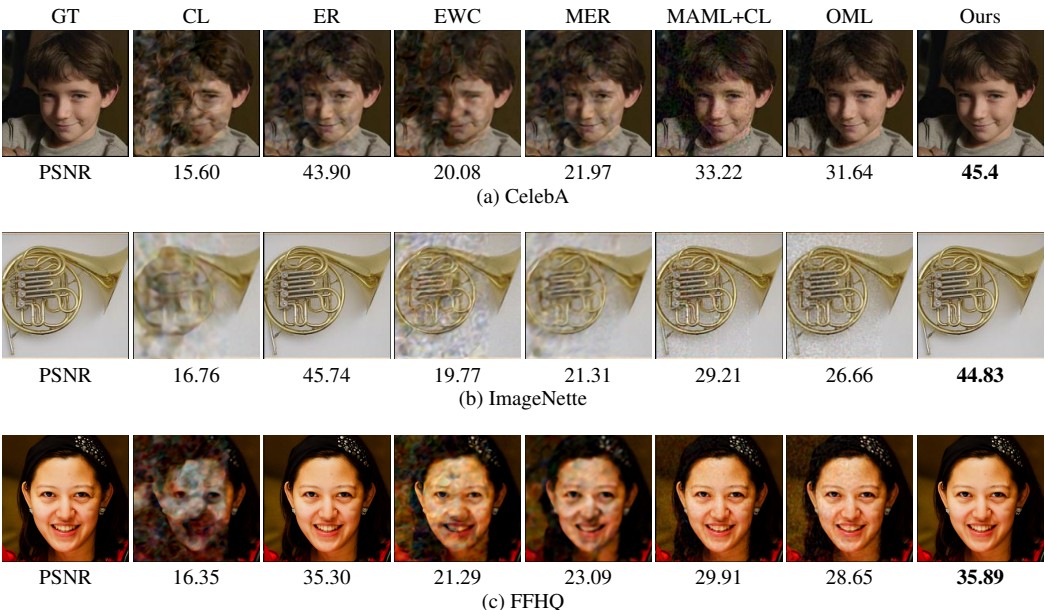

Figure 3: Qualitative results of image reconstruction. The first column represents the ground truth, while the remaining columns show the reconstruction results from different methods. The numbers below each image indicate the PSNR with respect to the ground truth. These images correspond to the Best results (those with the highest PSNR among 1 to 4096 steps), as presented in Table 1. For detailed results, please refer to Table 1.

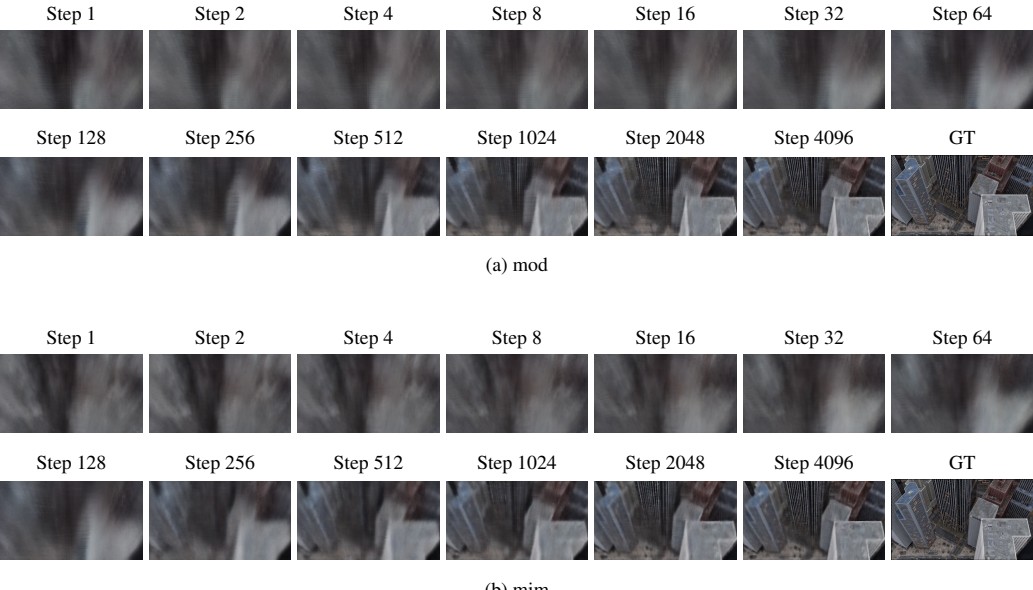

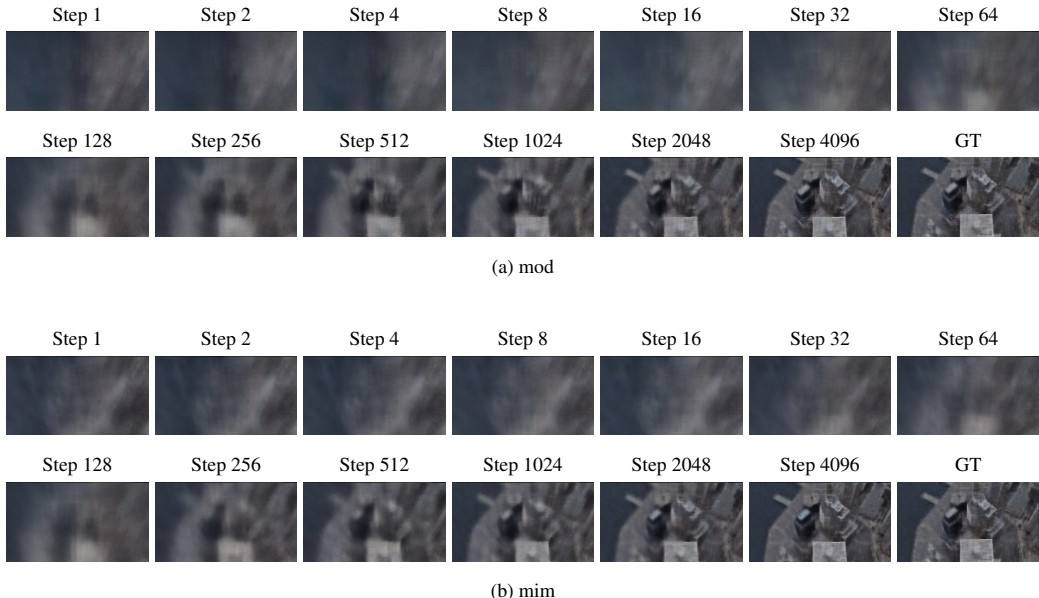

Figure 4: Visualization of Reconstruction Progression Over Steps for "mod" and "mim" Methods. (a) shows the progression of reconstruction for the "mod" (modularized) version, and (b) shows the "mim" (modularized with Fisher Information Maximization) version. Each row presents the reconstructed scene at different optimization steps (Step 1 to Step 4096) along with the ground truth (GT). The visualization demonstrates how the quality of reconstructed details gradually improves as the number of steps increases, highlighting the efficiency of both methods in learning the underlying structure of the scene, with "mim" exhibiting more rapid refinement.

