# OpenReview forum: "Meta-Continual Learning of Neural Fields"
_ICLR.cc/2025/Conference — ICLR 2025 Poster_

### Official Review · Reviewer_381t · 2024-11-02

**Soundness:** 3
**Presentation:** 2
**Contribution:** 2
**Rating:** 6
**Confidence:** 2

**Summary:**

In this paper, the authors propose a new problem setting and strategy to introduce meta-continual learning in neural fields. They conduct extensive experiments to demonstrate that their approach outperforms baseline methods.

**Strengths:**

1.	Experiments on various datasets are conducted to validate the effectiveness of the method.
2.	The method is general and could be applied to many different applications.

**Weaknesses:**

1.	The implementation and comparison are conducted based on methods published in 2021 and 2022. Only SwitchNeRF was published in ICLR 2023, which was also long ago. It is kindly suggested to conduct experiments on more recent methods to evaluate the proposed approach. I would like to see some comparative experiments with [a].
2.	I would expect more visualization in the experiments, for example, comparison of view synthesis on MatrixCity dataset.
3.	The experiments should include some commonly used evaluation metrics, such as SSIM.

[a] GF-NeRF: Global-guided Focal Neural Radiance Field for Large-scale Scene Rendering, WACV 2025

**Questions:**

See Weakness

---

> ### Author Response · Authors · 2024-11-25
>
> **Q. The implementation and comparison are conducted based on methods published in 2021 and 2022. Only SwitchNeRF was published in ICLR 2023, which was also long ago. It is kindly suggested to conduct experiments on more recent methods to evaluate the proposed approach. I would like to see some comparative experiments with [a].
> [a] GF-NeRF: Global-guided Focal Neural Radiance Field for Large-scale Scene Rendering, WACV 2025**
>
>
> Answer)
> In accordance with the reviewer's recommendation, we add [1] into our main table as an CL-NF method.
>
> | Modality |                    |             |              |           | NeRF       |           |            |        |
> |:----------:|:----------:|:----------:|:--------:|:------------:|:-----------:|:--------:|:------------:|:-----------:|
> | Dataset  |               |               |        | MatrixCity-B5     |           |        | MatrixCity-B6 |           |
> | Metric (PSNR) $\uparrow$ |  |  | Step 1 | Best (step) | Step 4096 | Step 1 | Best (step) | Step 4096 |
> | CL       | MEIL-NeRF[1] |   | 21.053 | 29.215 (1024) | 29.215  | 20.955  | 28.304 (1024) | 28.304 |
> | MCL   | Ours (mod)  |    | 23.885  | 32.712 (1024) | 32.712   | 23.217  | 30.407 (1024) | 30.407 |
> |          | Ours (MIM)  |    | **24.223**  | **32.804 (1024)** | **32.804**   | **23.341**  | **30.761 (1024)** | **30.761** |
>
> [1] MEIL-NeRF: Memory-Efficient Incremental Learning of Neural Radiance Fields. arXiv preprint arXiv:2212.08328, 2022.
>
> &nbsp;
>
> **Q. I would expect more visualization in the experiments, for example, comparison of view synthesis on MatrixCity dataset.**
>
>
> Answer) We appreciate the suggestion of visualization on NeRF. We have added visualization on the MatrixCity [1], which compares "Ours (mod)" and "Ours (MIM)", highlighting how the results change from step 1 to step 4096 in the 512 hidden dimensions of the neural network.
>
> [1] MatrixCity: A Large-scale City Dataset for City-scale Neural Rendering and Beyond. ICCV, 2023.

---

> ### Author Response · Authors · 2024-11-25
>
> **Q. The experiments should include some commonly used evaluation metrics, such as SSIM.**
>
> Answer)
> We calculate SSIM (Structural Similarity Index Measure) [2] for the image and video domains, and PESQ (Perceptual Evaluation of Speech Quality) [1] for the audio domain.
>
> The SSIM values show that our approach (Ours) maintains strong visual similarity, achieving higher scores at Step 4096 across all image and video datasets. Similarly, the PESQ values for the audio domain reveals that our method provides consistent perceptual quality compared to others, especially for longer adaptation sequences (Step 4096). That is, the robustness of our approach in quality is proven with both PSNR and perceptual quality metrics.
>
> | Modality |                    |             |              |           |  Image       |           |           |        |   |   Video |  |
> |:----------:|:---------------:|:---------------:|:--------:|:------------:|:-----------:|:--------:|:------------:|:-----------:|:--------:|:------------:|:-----------:|
> | Dataset  |               |               |        | CelebA     |           |        | ImageNette |           |        | VoxCeleb2       |           |
> | Metric (SSIM) $\uparrow$ |  |  | Step 1 | Best (step) | Step 4096 | Step 1 | Best (step) | Step 4096 | Step 1 | Best (step) | Step 4096 |****
> | OL       | OL            |               | 0.329  | 0.969 (4096) | 0.969  | 0.248 | 0.98 (4096)  | 0.98  | 0.347 | 0.967 (4096) | 0.967 |
> | CL       | CL            |               | 0.313  | 0.519 (64)   | 0.463  | 0.248 | 0.526 (128)  | 0.477 | 0.365 | 0.611 (64)   | 0.493 |
> |          | EWC           |               | 0.321  | 0.63 (256)   | 0.503  | 0.251 | 0.617 (256)  | 0.475 | 0.369 | 0.648 (256)  | 0.526 |
> | CML      | MER           |               | 0.332  | 0.711 (256)  | 0.633  | 0.251 | 0.674 (256)  | 0.581 | 0.348 | 0.68 (256)   | 0.603 |
> | MOL      | MOL           |               | 0.82   | **0.995 (4096)** | **0.995**  | 0.729 | **0.997 (4096)** | **0.997** | 0.807 | 0.991 (4096) | 0.991 |
> | MCL      | MAML+CL       |               | **0.822**  | 0.92 (64)    | 0.816  | 0.742 | 0.884 (32)   | 0.789 | 0.815 | 0.888 (32)   | 0.773 |
> |          | OML           |               | 0.796  | 0.884 (32)   | 0.65   | 0.696 | 0.81 (16)    | 0.553 | 0.796 | 0.855 (16)   | 0.599 |
> |          | Ours          |               | 0.817  | **0.995 (4096)** | **0.995**  | **0.751** | 0.996 (4096) | 0.996 | **0.857** | **0.996 (4096)** | **0.996** |
>
>
> | Modality |                    |              |           |  Audio       |           |
> |:----------:|:---------------:|:---------------:|:--------:|:------------:|:-----------:|
> | Dataset  |               |               |        | LibriSpeech1     |           |
> | Metric (PESQ) $\uparrow$ |  |  | Step 1 | Best (step) | Step 2048 |
> | OL       | OL            |    | 1.03 | 1.82 (2048) | 1.82 |
> | CL       | CL            |      | 1.02 | 1.12 (2048) | 1.12 |
> |          | ER           |        | 1.02 | 1.13 (4096)  | 1.13 |
> |          | EWC           |        | 1.02 | 1.12 (2048)  | 1.12 |
> | CML      | MER           |        | 1.03 | 1.16 (2048) | 1.16 |
> | MOL      | MOL           |          | 1.33 | 2.48 (4096) | 2.48 |
> | MCL      | MAML+CL  |        | **1.37** | 1.62 (32) | 1.59 |
> |          | OML           |               | 1.34 | 1.62 (512) | 1.59 |
> |          | Ours          |               | 1.3 | **3.73 (4096)** | **3.73** |
>
> [1] Perceptual evaluation of speech quality (PESQ)-a new method for speech quality assessment of telephone networks and codecs. ICASSP, 2001.
> [2] Image quality assessment: from error visibility to structural similarity. IEEE Image Process, 2004.
> [3] Learning to continually learn. ECAI, 2020.
> [4] La-MAML: Look-ahead Meta Learning for Continual Learning. NeurIPS, 2020.
> [5] Learning where to learn: Gradient sparsity in meta and continual learning.NeurIPS, 2021.
> [6] Meta Continual Learning Revisited: Implicitly Enhancing Online Hessian Approximation via Variance Reduction. ICLR, 2024.

---

### Official Review · Reviewer_R5BD · 2024-11-04

**Soundness:** 3
**Presentation:** 2
**Contribution:** 2
**Rating:** 6
**Confidence:** 3

**Summary:**

The paper introduces Meta-Continual Learning of Neural Fields (MCL-NF), a framework merging modular neural architectures with meta-learning strategies to support continuous and adaptive learning of neural fields. The proposed approach aims to address the challenges in traditional continual learning for neural fields, such as catastrophic forgetting and slow convergence. The authors implemented Fisher Information Maximization loss (FIM loss) to optimize learning by emphasizing more informative samples, achieving improved generalization and learning speed. The paper’s extensive experiments across various tasks including image, audio, video and NeRF-based view synthesis, highlight the performance advantages of MCL-NF over existing methods. They offer an efficient and scalable solution suitable for large-scale and resource-constrained environments, advancing the capabilities of neural fields in sequential learning settings.

**Strengths:**

1. MCL-NF’s combination of modular architecture and optimization based meta-learning is innovative and meets the adaptability needs of neural fields.
2. The method mitigates catastrophic forgetting through modularization and shared initialization, without the need for experience replay that can be resource-intensive. The modular approach within meta-continual learning sounds pretty compelling.
3. The FIM loss implementation is theoretically sound, supported by convergence guarantees, and introduces a detailed weighting mechanism for learning stability and efficiency.
4. The paper provides thorough experimental results, showcasing the model’s performance on various datasets, supporting the robustness of MCL-NF. The concept of evaluating the method in different modalities definitely deserves recognition.
5. This work helps understand the broader scope of how meta-learning techniques could be integrated successfully with frameworks for continual learning, particularly in the context of neural fields. This can help further research to consider similar synergies between learning paradigms, which may result in even more powerful and generalizable machine learning models.
6. The paper is well-organized and written in a clear, academic tone. It is suitably embedded in the meta-continual learning world  with relevant citations. The authors show their deep understanding of the topic by presenting the setting, highlighting its motivation and organizing the experiments in a structured way.

**Weaknesses:**

1. The setting, at least in the image domain, seems artificial - dividing the images into four patches (regardless of the size) seems to raise some concerns. It would be desirable to present an ablation study on it.
2. From a continual learning perspective, this setting is highly confusing. It should be explicitly described how the data is selected, processed and how the model is evaluated.
3. The paper lacks more recent methods in the main table. And even if MAML is a well-known method, it is quite outdated, with lots of recent enhancements such as La-MAML [1] for continual learning.  Moreover, the comparison is mostly made against methods in a different setting with only two methods in MCL (and MAML was published in 2017).
4. And even in different settings (not MCL), the papers are well-established, but some of them nowhere near the current state-of-the-art. There is no elaboration how this approach differs from recently presented settings/methods such as [2]. And even if they are mentioned, such as ANML they are not compared against.
5. Experimental results show the performance with a number of continual tasks set to only four, which raises concerns regarding the method’s scalability.
6. These experimental results are largely dependent on PSNR as the main metric of reconstruction quality. Additional metrics, such as Structural Similarity Index (SSIM) for image and video tasks, or perceptual evaluation metrics for audio, would provide a more comprehensive validation of MCL-NF’s performance.

Minor issues:
Line 155 - Redundant closing bracket

**Questions:**

1. Could additional strategies beyond Fisher Information, such as curriculum learning, improve MCL-NF's adaptability and reduce reliance on specific meta-learning techniques?
2. Should the paper also use different techniques of splitting the image than cropping into patches? The idea behind the method is based on that, and perhaps applying Fourier Transformations would be beneficial.
3. Could alternative loss functions achieve similar sample prioritization for continual learning? If so, what criteria would make FIM the preferred choice?
4. As far as I understand, we treat each image separately (batch size 1). How do we find the appropriate number of coordinates?
5. What insights can this setting bring to the continual learning community?
6. In the methods compared without modularization, do these approaches utilize the full MLP network capacity to match the parameter count of the modularized versions?
7. The Fisher Information Maximization loss introduces a new hyperparameter. Could you provide more details on how this hyperparameter was selected and discuss its sensitivity and impact on the model's performance?

---

> ### Author Response · Authors · 2024-11-25
>
> **Q. From a continual learning perspective, this setting is highly confusing. It should be explicitly described how the data is selected, processed and how the model is evaluated.**
>
> Answer)
> Thank you for highlighting this point. To clarify, our setting uses a Meta-Continual Learning framework which differs from traditional continual learning. In MCL, we focus on learning how to continually learn by employing a continual learner that takes old model parameters and new training data to update a model. This process involves both inner and outer loops: the inner loop refines model parameters using sequential data, while the outer loop optimizes meta-parameters to enhance the learning strategy.
> In the data selection and processing stage, the data are split into meta-training and meta-testing sets, each consisting of multiple tasks that simulate continual learning scenarios. These episodes are constructed to ensure that no tasks overlap between meta-training and meta-testing, promoting generalization. The data are sequentially provided to the model, where the model only sees new tasks as it progresses, similar to a traditional continual learning environment.
> Model Evaluation during meta-training, multiple episodes are used to optimize the meta-parameters through meta-gradient descent, ensuring the model can generalize well during meta-testing on unseen tasks. The performance is evaluated by assessing how well the model adapts to new tasks and retains knowledge from previous tasks.
>
> [1] Recasting Continual Learning as Sequence Modeling. NeurIPS, 2023.
> [2] When Meta-Learning Meets Online and Continual Learning: A Survey. TPAMI, 2024

---

> ### Author Response · Authors · 2024-11-25
>
> **Q. The paper lacks more recent methods in the main table. And even if MAML is a well-known method, it is quite outdated, with lots of recent enhancements such as La-MAML [1] for continual learning. Moreover, the comparison is mostly made against methods in a different setting with only two methods in MCL (and MAML was published in 2017).
> (+).
> Q. And even in different settings (not MCL), the papers are well-established, but some of them nowhere near the current state-of-the-art. There is no elaboration how this approach differs from recently presented settings/methods such as [2]. And even if they are mentioned, such as ANML they are not compared against.
> (+).
> Q. These experimental results are largely dependent on PSNR as the main metric of reconstruction quality. Additional metrics, such as Structural Similarity Index (SSIM) for image and video tasks, or perceptual evaluation metrics for audio, would provide a more comprehensive validation of MCL-NF’s performance.**
>
> Answer)
> We implement SSIM (Structural Similarity Index Measure) [1] for the image and video domains, and PESQ (Perceptual Evaluation of Speech Quality) [2] for the audio domain. Furthermore, we include ANML [6] as a newly implemented Meta-Continual Learning (MCL) method for comparison.
>
> In the table, we focus on a meta-continual learning specifically designed for learning to continually learn neural radiance fields.  We excluded the methods that are categorized as Continual Bi-level optimization [7, 8, 9] based on [10], since their settings differ from ours. MCL aims to create an initialization or strategy that quickly adapts to new tasks while retaining prior knowledge. In contrast, the latter focuses on adapting the meta-learner itself over time. Instead, we plan to experiment with multiple MAML variants [3, 4, 5] for the revision.
>
> The SSIM values show that our approach (Ours) maintains strong visual similarity, achieving higher scores at Step 4096 across all image and video datasets. Similarly, PESQ for the audio domain reveals that our method provides consistent perceptual quality compared to others, especially for longer adaptation sequences (Step 4096). Notably, our approach outperforms baseline models in terms of both PSNR and perceptual quality metrics, highlighting its robustness in maintaining quality during sequential learning.
>
> | Modality |                    |             |              |           |  Image       |           |           |        |   |   Video |  |
> |:----------:|:---------------:|:---------------:|:--------:|:------------:|:-----------:|:--------:|:------------:|:-----------:|:--------:|:------------:|:-----------:|
> | Dataset  |               |               |        | CelebA     |           |        | ImageNette |           |        | VoxCeleb2       |           |
> | Metric (SSIM) $\uparrow$ |  |  | Step 1 | Best (step) | Step 4096 | Step 1 | Best (step) | Step 4096 | Step 1 | Best (step) | Step 4096 |****
> | OL       | OL            |     | 0.329  | 0.969 (4096) | 0.969  | 0.248 | 0.98 (4096)  | 0.98  | 0.347 | 0.967 (4096) | 0.967 |
> | CL       | CL            |     | 0.313  | 0.519 (64)   | 0.463  | 0.248 | 0.526 (128)  | 0.477 | 0.365 | 0.611 (64)   | 0.493 |
> |          | EWC           |     | 0.321  | 0.63 (256)   | 0.503  | 0.251 | 0.617 (256)  | 0.475 | 0.369 | 0.648 (256)  | 0.526 |
> | CML      | MER       |    | 0.332  | 0.711 (256)  | 0.633  | 0.251 | 0.674 (256)  | 0.581 | 0.348 | 0.68 (256)   | 0.603 |
> | MOL      | MOL       |    | 0.82   | **0.995 (4096)** | **0.995**  | 0.729 | **0.997 (4096)** | **0.997** | 0.807 | 0.991 (4096) | 0.991 |
> | MCL      | MAML+CL       |               | **0.822**  | 0.92 (64)    | 0.816  | 0.742 | 0.884 (32)   | 0.789 | 0.815 | 0.888 (32)   | 0.773 |
> |          | OML           |               | 0.796  | 0.884 (32)   | 0.65   | 0.696 | 0.81 (16)    | 0.553 | 0.796 | 0.855 (16)   | 0.599 |
> |          | Ours          |               | 0.817  | **0.995 (4096)** | **0.995** | **0.751** | 0.996 (4096) | 0.996 | **0.857** | **0.996 (4096)** | **0.996** |
>
> | Modality |       |       |           |  Audio       |           |
> |:----------:|:---------------:|:---------------:|:--------:|:------------:|:-----------:|
> | Dataset  |               |               |        | LibriSpeech1     |           |
> | Metric (PESQ) $\uparrow$ |  |  | Step 1 | Best (step) | Step 2048 |
> | OL       | OL            |    | 1.03 | 1.82 (2048) | 1.82 |
> | CL       | CL            |      | 1.02 | 1.12 (2048) | 1.12 |
> |          | ER           |        | 1.02 | 1.13 (4096)  | 1.13 |
> |          | EWC           |        | 1.02 | 1.12 (2048)  | 1.12 |
> | CML      | MER           |        | 1.03 | 1.16 (2048) | 1.16 |
> | MOL      | MOL           |          | 1.33 | 2.48 (4096) | 2.48 |
> | MCL      | MAML+CL  |        | **1.37** | 1.62 (32) | 1.59 |
> |          | OML           |               | 1.34 | 1.62 (512) | 1.59 |
> |          | Ours          |               | 1.3 | **3.73 (4096)** | **3.73** |

---

> ### Author Response · Authors · 2024-11-25
>
> [1] Perceptual evaluation of speech quality (PESQ)-a new method for speech quality assessment of telephone networks and codecs. ICASSP, 2001.
> [2] Image quality assessment: from error visibility to structural similarity. IEEE Image Process, 2004.
> [3] Meta-SGD: Learning to Learn Quickly for Few Shot Learning. arXiv preprint arXiv:1707.09835, 2017.
> [4] On First-Order Meta-Learning Algorithms. arXiv preprint arXiv:1803.02999, 2018.
> [5] How to train your MAML. ICLR 2019.
> [6] Learning to continually learn. ECAI, 2020.
> [7] La-MAML: Look-ahead Meta Learning for Continual Learning. NeurIPS, 2020.
> [8] Learning where to learn: Gradient sparsity in meta and continual learning. NeurIPS, 2021.
> [9] Meta Continual Learning Revisited: Implicitly Enhancing Online Hessian Approximation via Variance Reduction. ICLR, 2024.
> [10] When Meta-Learning Meets Online and Continual Learning: A Survey. TPAMI, 2024.
>
>   &nbsp;
>
> **Q. Experimental results show the performance with a number of continual tasks set to only four, which raises concerns regarding the method’s scalability.**
>
> Answer)
> We present the results of comparison between Ours (mod) and Ours (MIM) with two different numbers of continual tasks (5 and 10) in Table below. The results demonstrate that MIM consistently outperforms the counterpart mod, which is consistent with the results from other settings. It is worth mentioning that the improvement gap is not as signiìcant as in other settings, likely due to limited hyperparameter tuning and training time available during the rebuttal period. We plan to conduct extensive experiments on longer sequences and will include these results in the revision.
>
> |# of tasks | Method | Step 1 | Step 2 | Step 4 | Step 8 | Step 16 | Step 32 | Step 64 | Step 128 | Step 256 | Step 512 | Step 1024 |
> | :--------: | :--------: | :--------: | :--------: | :--------: | :--------: | :--------: | :--------: | :--------: | :--------: | :--------: | :--------: | :--------: |
> | 5 | **Ours (mod)** | _24.072_  | _24.155_ | **_24.484_** | _24.696_ | _25.072_  | **_25.631_** | _26.645_ | _28.075_  | _29.665_ | _30.836_ | _31.681_  |
> | | **Ours (MIM)** | **_24.143_**  | **_24.276_**  | _24.394_ | **_24.775_**  | **_25.102_**  | _25.61_ | **_26.83_**  | **_28.23_**  | **_29.73_**  | **_31.005_**  | **_31.822_**  |
> | 10 | **Ours (mod)** | _23.415_  | _23.514_ | _23.618_ | _24.001_  | _24.298_ | _24.768_ | _26.037_  | _27.62_ | _28.88_ | **_29.76_**  | _31.068_  |
> | | **Ours (MIM)** | **_23.493_**  | **_23.548_**  | **_23.783_**  | **_24.119_**  | **_24.464_**  | **_24.793_**  | **_26.238_**  | **_27.667_**  | **_28.886_** | _29.71_  | **_31.098_**  |
>
> **Table:** Performance metrics (PSNR) for "Ours (mod)" and "Ours (MIM)" across different optimization steps, for 5 and 10 tasks. The metrics are evaluated at steps doubling from 1 to 1024.
>
> &nbsp;
>
> **Q. Could additional strategies beyond Fisher Information, such as curriculum learning, improve MCL-NF's adaptability and reduce reliance on specific meta-learning techniques?**
>
> Answer)
> Incorporating additional strategies like curriculum learning could further improve MCL-NF's adaptability while reducing its reliance on specific meta-learning techniques. As discussed in [3] curriculum learning systematically orders training samples from easier to harder tasks, mimicking the natural learning process. This progressive approach could help MCL-NF transition more smoothly between tasks [2], improve convergence[1], or adapt to complex scenes [4].
>
> [1] Curriculum Meta Learning: Learning to Learn from Easy to Hard. EITCE, 2021.
> [2] Curriculum Meta-Learning for Few-shot Classification. arXiv preprint arXiv2112.02913, 2021.
> [3] Curriculum Learning: A Survey. IJCV, 2023.
> [4] Learning Large-scale Neural Fields via Context Pruned Meta-Learning. NeurIPS, 2023.

---

> ### Author Response · Authors · 2024-11-25
>
> **Q. Could alternative loss functions achieve similar sample prioritization for continual learning? If so, what criteria would make FIM the preferred choice?**
>
> Answer)
> Yes, alternative loss functions like bootstrap-based ones (e.g., GradNCP [1]) can also achieve sample prioritization for continual learning by dynamically selecting context points that promise high immediate gains in quality. GradNCP, for example, uses a bootstrap correction to focus on the data that minimize the error. However, Fisher Information Maximization (FIM) stands out for large-scale NeRF settings, especially where computational resources are constrained.
> FIM inherently measures the importance of model parameters relative to each data sample, effectively prioritizing those with the greatest impact on parameter changes. This makes FIM particularly efficient for managing memory and computation in large-scale settings. Unlike bootstrap methods, FIM also helps mitigate catastrophic forgetting by preserving parameters crucial for prior tasks, striking a balance between learning new information and retaining essential knowledge.
> Also, we incorporate GradNCP into the table below for comparison.
>
> | Method |             |              |           |  NeRF       |           |           |        |
> |:----------:|:---------------:|:--------:|:------------:|:-----------:|:--------:|:------------:|:-----------:|
> | Dataset  |               |        | MatrixCity-B5     |           |        | MatrixCity-B6 |           |
> | Metric (PSNR) $\uparrow$ || Step 1 | Best (step) | Step 1024 | Step 1 | Best (step) | Step 1024 |
> | GradNCP  |             | 21.256  | 27.815 (1024) | 27.815   | 21.655  | 26.156 (1024) | 26.156  |
> | Ours      |                 | **24.223**  | **32.804 (1024)**  | **32.804**   | **23.341**  | **30.761 (1024)**   | **30.761**   |
>
> **Table:** Comparison between sample prioritization methods.
>
> [1] Learning Large-scale Neural Fields via Context Pruned Meta-Learning. NeurIPS, 2023.
>
> &nbsp;
>
> **Q. As far as I understand, we treat each image separately (batch size 1). How do we find the appropriate number of coordinates?**
>
> Answer)
> Increasing the number of coordinates generally leads to better performance, since it allows the model to capture more details from the scene. When handling complex, large-scale scenes, it is advisable to select as many coordinates as computational resources allow, for improving the quality of the final reconstruction.
> For determining the appropriate number of coordinates, it is helpful to look at prior works such as Mega-NeRF[1] and Switch-NeRF[2], which tackle large-scale scenes and employ a large number of sampled rays (typically around 1024). However, in our setting for meta-continual learning in NeRF, we opted to reduce the number of sampled coordinates to 256—considerably smaller than in Mega-NeRF. This reduction helps reduce computational time and memory usage, which are important for the continual learning constraints and the need for efficient processing.
> In summary, it is important to acknowledge that the number of coordinates should be chosen based on the trade-off between available computational resources and the desired reconstruction quality.
>
> [1] Mega-NeRF: Scalable Construction of Large-Scale NeRFs for Virtual Fly-Throughs, CVPR 2022.
> [2] Switch-NeRF: Learning Scene Decomposition with Mixture of Experts for Large-scale Neural Radiance Fields, ICLR 2023.

---

> ### Author Response · Authors · 2024-11-25
>
> **Q. What insights can this setting bring to the continual learning community?**
>
> Answer)
> The key insights of the setting is to demonstrate how modularization and neural field representations can address the issues like catastrophic forgetting and slow convergence in complex, continuous environments. Specifically, it shows that neural networks can serve as efficient memory, compressing scene representations with no explicit data storage, which is useful for continual adaptation. The application of continual learning to continuous radiance fields illustrates how spatial continuity can be handled incrementally, bridging the gap between traditional task-based learning and the need for adaptive scene understanding in 3D modeling. These insights may help develop new strategies for retaining and adapting knowledge efficiently in dynamic, large-scale environments.
>
> &nbsp;
>
> **Q. In the methods compared without modularization, do these approaches utilize the full MLP network capacity to match the parameter count of the modularized versions?**
>
> Answer)
> To ensure a fair and controlled comparison, the parameter counts of the modularized and un-modularized versions are matched by adjusting the hyperparameters of the model size.
>
> | Method | Number of Networks | Hidden Dimension | Number of Layers | Parameter Count (with Bias) |
> |:-------------------:|:--------------------:|:------------------:|:------------------:|:---------------------:|
> | Un-modularized    | 1                  | 512              | 10               | 2,628,099                     |
> | Modularized       | 4                  | 256              | 10               | 4 x 659,459 = 2,637,836 |
>
> **Table:** Comparison of parameter counts for different network configurations, including un-modularized and modularized versions with bias terms included.
>
> &nbsp;
>
> **Q. The implementation and comparison are conducted based on methods published in 2021 and 2022. Only SwitchNeRF was published in ICLR 2023, which was also long ago. It is kindly suggested to conduct experiments on more recent methods to evaluate the proposed approach. I would like to see some comparative experiments with [a].
> [a] GF-NeRF: Global-guided Focal Neural Radiance Field for Large-scale Scene Rendering, WACV 2025**
>
> Answer)
> In accordance with the reviewer's recommendation, we add [1] into our main table as an CL-NF method.
>
> | Modality |                    |             |              |           | NeRF       |           |            |        |
> |:----------:|:----------:|:----------:|:--------:|:------------:|:-----------:|:--------:|:------------:|:-----------:|
> | Dataset  |               |               |        | MatrixCity-B5     |           |        | MatrixCity-B6 |           |
> | Metric (PSNR) $\uparrow$ |  |  | Step 1 | Best (step) | Step 1024 | Step 1 | Best (step) | Step 1024 |
> | CL       | MEIL-NeRF[1] |   | 21.053 | 29.215 (1024) | 29.215  | 20.955  | 28.304 (1024) | 28.304 |
> | MCL   | Ours (mod)  |    | 23.885  | 32.712 (1024) | 32.712   | 23.217  | 30.407 (1024) | 30.407 |
> |          | Ours (MIM)  |    | **24.223**  | **32.804 (1024)** | **32.804**   | **23.341**  | **30.761 (1024)** | **30.761**   |
>
> [1] MEIL-NeRF: Memory-Efficient Incremental Learning of Neural Radiance Fields. arXiv preprint arXiv:2212.08328, 2022.

---

> > ### Comment · Reviewer_R5BD · 2024-11-26
> >
> > Thanks for your answers. The results of additional experiments answer my questions. Please include these experiments in the paper.

---

> ### Author Response · Authors · 2024-11-27
>
> **Q. The setting, at least in the image domain, seems artificial - dividing the images into four patches (regardless of the size) seems to raise some concerns. It would be desirable to present an ablation study on it.**
>
> Answer)
> We initially experimented with dividing images using the Fourier Transform instead of patches. However, handling complex numbers did not integrate well with our pipeline. As a result, we are now focusing on dividing images into lower resolutions. For instance, an n x n image can be split into four (n//2) x (n//2) images (in same height and width), which we then optimize sequentially.
>
> | Modality   |                 |              |              |             | Image        |            |            |          |    Video       |      |
> |:----------:|:---------------:|:------------:|:------------:|:-----------:|:------------:|:----------:|:----------:|:--------:|:-----------:|:-----------:|
> | Dataset    |                 |              |            CelebA  |     |       |     ImageNette |      |    | VoxCeleb2   |
> | Metric (PSNR) $\uparrow$ |    | Step 1 | Best (step) | Step 4096 | Step 1 | Best (step) | Step 4096 | Step 1 | Best (step) | Step 4096 |
> | OL         | OL              |       11.84 | 38.2 (4096)  | 38.2  | 12.19  | 38.87 (4096)  | 38.87  | 13  | 37.83 (4096)  | 37.83 |
> | CL         | CL              |        17.63  | 26.3 (4096)    | 26.3  | 16.7  | 26.67 (2048)  | 26.67  | 17.53  | 33.77 (1024)    | 33.39 |
> |           | ER             |         17.63  | 26.5 (1024)    | 26.47  | 16.7  | 26.69 (2048)  | 26.66  | 17.53  | 33.79 (1024)   | 33.44 |
> |           | EWC             |        18.21  | 28.12 (4096)    | 28.12  | 17.01  | 26.88 (1024)  | 26.75  | 17.96  | 34.18 (4096)   | 34.18 |
> | CML        | MER             |       12.21  | 26.62 (4096)   | 26.62  | 12.56  | 26.62 (2048)  | 26.61  | 13.03  | 33.94 (4096)    | 33.94 |
> | MOL        | MOL            |        29.28   | 46.36 (4096) | 46.36  | 24.52  | **48.55 (4096)** | **48.55**  | 28.9  | **44.21 (4096)**  | **44.21** |
> | MCL        | MAML+CL      |  **33.12**  | 34.12 (4)  | 32.6  | **27.6**  | 28.24 (2)   | 26.83  | 33.65  | 36.3 (16)    | 34.09 |
> |           | OML             |       32.55  | 33.69 (4)    | 33.14   | 26.96  | 27.53 (2)    | 26.85  | **34.05**  | 36.67 (16)    | 35.19 |
> |           | Ours            | 24.22  | **48.94 (4096)** | **48.94** | 20.62 | 46.81 (4096)  | 46.81  | 24.81  | 42.31 (4096) | 42.31 |
>
> | Modality   |                 |              |              |        Image  |        |            |            |        |    Video       |   |
> |:----------:|:---------------:|:------------:|:------------:|:-------:|:------------:|:----------:|:----------:|:--------:|:-----------:|:-----------:|
> | Dataset    |                 |              |          CelebA   |    |            | ImageNette |      |   |  VoxCeleb2  |   |
> | Metric (SSIM) $\uparrow$ |   | Step 1 | Best (step) | Step 4096 | Step 1 | Best (step) | Step 4096 | Step 1 | Best (step) | Step 4096 |
> | OL         | OL                          | 0.329  | 0.969 (4096)  | 0.969  | 0.248  | 0.98 (4096)  | 0.98  | 0.347  | 0.967 (4096)  | 0.967 |
> | CL         | CL                          | 0.469  | 0.915 (4096)    | 0.915  | 0.344  | 0.892 (4096)  | 0.892  | 0.438  | 0.93 (1024)    | 0.918 |
> |           | ER                          | 0.469  | 0.916 (4096)    | 0.916  | 0.344  | 0.893 (2048)  | 0.892  | 0.438  | 0.93 (1024)   | 0.919 |
> |           | EWC                         | 0.483 | 0.926 (4096)    | 0.926  | 0.357  | 0.892 (2048)  | 0.891  | 0.462  | 0.935 (2048)   | 0.932 |
> | CML        | MER                  | 0.343  | 0.914 (4096)   | 0.914  | 0.258  | 0.882 (4096)  | 0.882  | 0.356  | 0.93 (4096)    | 0.93 |
> | MOL        | MOL    | 0.82   | 0.995 (4096) | 0.995  | 0.729  | **0.997 (4096)** | **0.997**  | 0.807  | **0.991 (4096)**  | **0.991** |
> | MCL        | MAML+CL             | **0.912**  | 0.933 (64)    | 0.902  | **0.852**  | 0.878 (32)   | 0.826  | **0.913** | 0.95 (16)  | 0.912 |
> |           | OML                  | 0.892  | 0.925 (8)    | 0.915   | 0.805  | 0.836 (4)    | 0.822  | **0.913**  | 0.952 (16)    | 0.925 |
> |           | Ours       | 0.669  | **0.996 (4096)** | **0.996** | 0.532 | 0.996 (4096)  | 0.996  | 0.659  | 0.984 (4096) | 0.984 |

---

### Official Review · Reviewer_6fGr · 2024-11-04

**Soundness:** 3
**Presentation:** 3
**Contribution:** 2
**Rating:** 6
**Confidence:** 4

**Summary:**

The problem of continual learning in neural fields is an important topic. This work focuses on addressing the issues of catastrophic forgetting and slow convergence in existing approaches by proposing a new paradigm called MCL-NF. It introduces the Fisher Information Maximization loss. The experiments demonstrate that the proposed method can achieve satisfactory results.

**Strengths:**

1.	This paper is well-written.
2.	This work combines meta-learning and continual learning paradigms into neural field training to address the challenges of catastrophic forgetting and slow convergence, which is interesting.
3.	This paper demonstrates the advantages of their method in experiments, which alleviates the issue of slow convergence to some extent.

**Weaknesses:**

1.	In the experimental section, the limited scenarios are my main concern, which makes me worry about the limitations of the method.
2.	The paper uses a large number of quantitative metrics but lacks qualitative comparisons, especially for neural radiance fields.
3.	In neural fields, some incremental learning methods [1] might also need to be discussed, but they are not mentioned in the article.

[1] Zhang, L., Li, M., Chen, C., & Xu, J. (2023). IL-NeRF: Incremental Learning for Neural Radiance Fields with Camera Pose Alignment. arXiv preprint arXiv:2312.05748.

4.  The paper is recommended to discuss the differences between continuous radiance fields and traditional continual learning. You know, issues like catastrophic forgetting and slow convergence have long been a focus of continual learning researchers.

**Questions:**

Please kindly refer to the above weaknesses.

---

> ### Author Response · Authors · 2024-11-25
>
> **Q. The paper uses a large number of quantitative metrics but lacks qualitative comparisons, especially for neural radiance fields.**
>
> Answer)
> We thank the reviewer for the insight! We have incorporated visualization on the MatrixCity [1] dataset to improve reader comprehension. The visualization compares "Ours (mod)" and "Ours (MIM)", highlighting how the results change from step 1 to step 4096 in the setting of a 512 hidden dimension size of the neural network.
>
> [1] MatrixCity: A Large-scale City Dataset for City-scale Neural Rendering and Beyond. ICCV, 2023.
>
> &nbsp;
>
> **Q. In neural fields, some incremental learning methods might also need to be discussed, but they are not mentioned in the article.**
>
> Answer)
> In accordance with the reviewer's recommendation, we incorporate [1] into our main table as a CL-NF method. (We will include [2] in the revision.) Both methods use self-distillation to mitigate catastrophic forgetting, leading to performance improvements over optimization steps. However, they require significantly more optimization to achieve comparable results and often still fall short of our method in reconstruction quality.
>
> | Modality |                    |             |              |           | NeRF       |           |            |        |
> |:----------:|:----------:|:----------:|:--------:|:------------:|:-----------:|:--------:|:------------:|:-----------:|
> | Dataset  |               |               |        | MatrixCity-B5     |           |        | MatrixCity-B6 |           |
> | Metric (PSNR) $\uparrow$ |  |  | Step 1 | Best (step) | Step 1024 | Step 1 | Best (step) | Step 1024 |
> | CL       | MEIL-NeRF[1] |   | 21.053 | 29.215 (1024) | 29.215  | 20.955  | 28.304 (1024) | 28.304 |
> | MCL   | Ours (mod)  |    | 23.885  | 32.712 (1024) | 32.712   | 23.217  | 30.407 (1024) | 30.407 |
> |          | Ours (MIM)  |    | **24.223**  | **32.804 (1024)** | **32.804**   | **23.341**  | **30.761 (1024)** | **30.761**   |
>
> [1] MEIL-NeRF: Memory-Efficient Incremental Learning of Neural Radiance Fields. arXiv preprint arXiv:2212.08328, 2022.
> [2] Instant Continual Learning of Neural Radiance Fields. ICCVW, 2023.

---

> ### Author Response · Authors · 2024-11-25
>
> **Q. The paper is recommended to discuss the differences between continuous radiance fields and traditional continual learning. You know, issues like catastrophic forgetting and slow convergence have long been a focus of continual learning researchers.**
>
> Answer)
> *Thank you for the insightful recommendation. We will add the following in our draft.*
>
> Continual Learning (CL) and Continual Learning for Neural Radiance Fields (CL-NeRF) [1, 4, 5] differ in both their goals and the challenges they address. In traditional CL, the objective is to train a model on a sequence of distinct tasks while mitigating catastrophic forgetting—the phenomenon where learning a new task leads to the deterioration of performance on previous tasks. CL-NeRF, on the other hand, focuses on learning and adapting 3D scene representations over time. Instead of distinct tasks, the challenge here involves incremental learning of a continuous environment, such as parts of a large city [2, 3]  or evolving scenes [6]. In CL-NeRF, a neural network serves as a memory, which is fundamentally different from traditional replay buffers in CL. By directly encoding scene geometry and appearance into network parameters, the network acts as a compression mechanism, reducing the need for storing explicit images or 3D points. This enables the representation of vast amounts of scene data in a compact, parameterized form.
> The concept of forgetting also differs between the two settings. In traditional CL, catastrophic forgetting impacts task-specific performance metrics (e.g., classification accuracy). In CL-NeRF, forgetting translates into the inability to render previously learned parts of a 3D scene accurately, resulting in visual artifacts or missing details. The continuity of the scene means that forgetting cannot be isolated to a single "task"—it affects the overall coherence of the rendered environment.
> Furthermore, slow convergence is another key challenge in both fields but manifests differently. In traditional CL, slow convergence may hinder effective adaptation across a sequence of tasks, while in CL-NF, it delays rendering a high-quality representation of a dynamic scene. Since NeRF must balance learning new features of a scene without degrading the quality of previously learned areas, computational efficiency becomes paramount.
> In summary, while both traditional CL and CL-NF aim to manage incremental learning, their key differences lie in the form of memory (explicit replay vs. network parameterization), the implications of forgetting (task-specific vs. continuous scene degradation), and the nature of learning targets (discrete tasks vs. spatially continuous environments). CL-NF uniquely uses neural networks not only to learn but also to compress scene information, enabling efficient use of memory and improving adaptability to evolving environments.
>
> [1] MEIL-NeRF: Memory-Efficient Incremental Learning of Neural Radiance Fields. arXiv preprint arXiv:2212.08328, 2022.
> [2] Capturing, Reconstructing, and Simulating: the UrbanScene3D Dataset. ECCV, 2022.
> [3] CityNeRF: Building NeRF at City Scale. ECCV, 2022.
> [4] Instant Continual Learning of Neural Radiance Fields. ICCVW, 2023.
> [5] IL-NeRF: Incremental Learning for Neural Radiance Fields with Camera Pose Alignment. arXiv preprint arXiv:2312.05748, 2023.
> [6] CL-NeRF: Continual Learning of Neural Radiance Fields for Evolving Scene Representation. NeurIPS, 2023.

---

> ### Author Response · Authors · 2024-11-27
>
> **Q. In the experimental section, the limited scenarios are my main concern, which makes me worry about the limitations of the method.**.
>
> Answer)
> We initially experimented with dividing images using the Fourier Transform instead of patches. However, handling complex numbers did not integrate well with our pipeline. As a result, we are now focusing on dividing images into lower resolutions. For instance, an n x n image can be split into four (n//2) x (n//2) images (in same height and width), which we then optimize sequentially.
>
> The newly experimented comparison aligns with previous results, except that the upper bound baseline rarely outperforms ours because our approach benefits from faster adaptation through meta-initializations, allowing for better generalization across tasks.
>
> | Modality   |                 |              |              |             | Image        |            |            |          |    Video       |      |
> |:----------:|:---------------:|:------------:|:------------:|:-----------:|:------------:|:----------:|:----------:|:--------:|:-----------:|:-----------:|
> | Dataset    |                 |              |            CelebA  |     |       |     ImageNette |      |    | VoxCeleb2   |
> | Metric (PSNR) $\uparrow$ |    | Step 1 | Best (step) | Step 4096 | Step 1 | Best (step) | Step 4096 | Step 1 | Best (step) | Step 4096 |
> | OL         | OL              |       11.84 | 38.2 (4096)  | 38.2  | 12.19  | 38.87 (4096)  | 38.87  | 13  | 37.83 (4096)  | 37.83 |
> | CL         | CL              |        17.63  | 26.3 (4096)    | 26.3  | 16.7  | 26.67 (2048)  | 26.67  | 17.53  | 33.77 (1024)    | 33.39 |
> |           | ER             |         17.63  | 26.5 (1024)    | 26.47  | 16.7  | 26.69 (2048)  | 26.66  | 17.53  | 33.79 (1024)   | 33.44 |
> |           | EWC             |        18.21  | 28.12 (4096)    | 28.12  | 17.01  | 26.88 (1024)  | 26.75  | 17.96  | 34.18 (4096)   | 34.18 |
> | CML        | MER             |       12.21  | 26.62 (4096)   | 26.62  | 12.56  | 26.62 (2048)  | 26.61  | 13.03  | 33.94 (4096)    | 33.94 |
> | MOL        | MOL            |        29.28   | 46.36 (4096) | 46.36  | 24.52  | **48.55 (4096)** | **48.55**  | 28.9  | **44.21 (4096)**  | **44.21** |
> | MCL        | MAML+CL      |  **33.12**  | 34.12 (4)  | 32.6  | **27.6**  | 28.24 (2)   | 26.83  | 33.65  | 36.3 (16)    | 34.09 |
> |           | OML             |       32.55  | 33.69 (4)    | 33.14   | 26.96  | 27.53 (2)    | 26.85  | **34.05**  | 36.67 (16)    | 35.19 |
> |           | Ours            | 24.22  | **48.94 (4096)** | **48.94** | 20.62 | 46.81 (4096)  | 46.81  | 24.81  | 42.31 (4096) | 42.31 |
>
> | Modality   |                 |              |              |        Image  |        |            |            |        |    Video       |   |
> |:----------:|:---------------:|:------------:|:------------:|:-------:|:------------:|:----------:|:----------:|:--------:|:-----------:|:-----------:|
> | Dataset    |                 |              |          CelebA   |    |            | ImageNette |      |   |  VoxCeleb2  |   |
> | Metric (SSIM) $\uparrow$ |   | Step 1 | Best (step) | Step 4096 | Step 1 | Best (step) | Step 4096 | Step 1 | Best (step) | Step 4096 |
> | OL         | OL                          | 0.329  | 0.969 (4096)  | 0.969  | 0.248  | 0.98 (4096)  | 0.98  | 0.347  | 0.967 (4096)  | 0.967 |
> | CL         | CL                          | 0.469  | 0.915 (4096)    | 0.915  | 0.344  | 0.892 (4096)  | 0.892  | 0.438  | 0.93 (1024)    | 0.918 |
> |           | ER                          | 0.469  | 0.916 (4096)    | 0.916  | 0.344  | 0.893 (2048)  | 0.892  | 0.438  | 0.93 (1024)   | 0.919 |
> |           | EWC                         | 0.483 | 0.926 (4096)    | 0.926  | 0.357  | 0.892 (2048)  | 0.891  | 0.462  | 0.935 (2048)   | 0.932 |
> | CML        | MER                  | 0.343  | 0.914 (4096)   | 0.914  | 0.258  | 0.882 (4096)  | 0.882  | 0.356  | 0.93 (4096)    | 0.93 |
> | MOL        | MOL    | 0.82   | 0.995 (4096) | 0.995  | 0.729  | **0.997 (4096)** | **0.997**  | 0.807  | **0.991 (4096)**  | **0.991** |
> | MCL        | MAML+CL             | **0.912**  | 0.933 (64)    | 0.902  | **0.852**  | 0.878 (32)   | 0.826  | **0.913** | 0.95 (16)  | 0.912 |
> |           | OML                  | 0.892  | 0.925 (8)    | 0.915   | 0.805  | 0.836 (4)    | 0.822  | **0.913**  | 0.952 (16)    | 0.925 |
> |           | Ours       | 0.669  | **0.996 (4096)** | **0.996** | 0.532 | 0.996 (4096)  | 0.996  | 0.659  | 0.984 (4096) | 0.984 |

---

> ### Author Response · Authors · 2024-11-29
>
> *We have provided additional answers to further supplement our previous responses.*
>  &nbsp;
>
> **Q. In neural fields, some incremental learning methods [1] might also need to be discussed, but they are not mentioned in the article.**
>
> Thank you for suggesting the inclusion of MEIL-NeRF [1], ICL-NeRF [2], and IL-NeRF [3] in the related work. As these works are closed-source, we are unable to conduct a formal quantitative comparison at this time. However, we will discuss the differences between these methods and ours, with an informal comparison to the MEIL-NeRF experiment provided response above.
>
> >**Comparison to *MEIL-NeRF*, *ICL-NeRF*, and *IL-NeRF*:**
> MEIL-NeRF [1], ICL-NeRF [2], and IL-NeRF [3] differ from our approach in their frameworks. While they utilize continual learning techniques for neural fields, they primarily rely on self-distillation to mitigate catastrophic forgetting, which, although effective, can impede the overall optimization process. In contrast, our method leverages meta-initializations, enabling fast adaptation and requiring only a few adaptation steps, which significantly accelerates learning and reduces the optimization burden. Additionally, MEIL-NeRF, ICL-NeRF, and IL-NeRF depend on either implicit or explicit memory mechanisms to handle forgetting, whereas our approach eliminates the need for such memory requirements, ensuring more efficient and resource-friendly optimization without sacrificing performance.
>
> [1] MEIL-NeRF: Memory-Efficient Incremental Learning of Neural Radiance Fields. arXiv preprint arXiv:2212.08328, 2022.
> [2] Instant Continual Learning of Neural Radiance Fields. ICCVW, 2023.
> [3] IL-NeRF: Incremental Learning for Neural Radiance Fields with Camera Pose Alignment. arXiv preprint arXiv:2312.05748.
> &nbsp;
>
> Please let us know if you have any further questions, and we are happy to incorporate additional suggestions you might have!
>
> If you find our response satisfactory, we would be grateful if you could consider raising your score.
>
> Thanks again for your time and efforts!

---

### Official Review · Reviewer_pV54 · 2024-11-05

**Soundness:** 3
**Presentation:** 3
**Contribution:** 3
**Rating:** 6
**Confidence:** 4

**Summary:**

Briefly, this paper presents a strategy to continually and rapidly learn neural fields in a meta-learning manner. The paper further introduces a fisher information maximization loss for neural radiance fields. The proposed method advances in leading to no performance degradation incurred by forgetting during test-time by synergizing modular architecture with meta-learning.

**Strengths:**

+ The paper is well written and easy to follow.
+ Extensive and comprehensive experiments demonstrate the effectiveness of the proposed method.

**Weaknesses:**

+ The technical novelty of the proposed method seems to be marginal since the authors directly employ the existing techniques. For instance, Fisher Information against catastrophic forgetting has already been proposed by Kirkpatrick et al., 2016. The authors do discuss the relation to this method and claim that their proposed FIM loss operates at the sample level rather than the parameter level. However, it is difficult to identify the advantage of the proposed FIM loss over the parameter-level approaches in CL (Chaudhry et al., 2018; Konishi et al., 2023) without fair experimental comparison and detailed discussion.

+ More detailed discussions and analyses in Table 1 are required to demonstrate the contribution of the proposed method.

+ More recent state-of-the-art methods should be included for comparison to demonstrate the superiority of the proposed method. For instance, missing some SOTA methods, e.g., (Chung et al., 2022 and Po et al., 2023), for comparison on the MatrixCity dataset in Table 1.

+ More ablation studies in the main paper are required to demonstrate the contribution of the main component (i.e., the FIM loss) of the proposed method in the main paper. The authors only provide some results about the mod and MIM  with two different hidden dimensions in Appendix Table 2.

Reference:
Po et al., Instant Continual Learning of Neural Radiance Fields, In ICCVW 2023.

**Questions:**

Please refer to the Weaknesses section.

There is a wrong citation in either Line 322 (EWC Chaudhry et al., 2018) or Line 402 (EWC (Kirkpatrick et al., 2016) ). Actually, Chaudhry et al., 2018 denote their method as EWC++.

---

> ### Author Response · Authors · 2024-11-25
>
> **Q. The technical novelty of the proposed method seems to be marginal since the authors directly employ the existing techniques. For instance, Fisher Information against catastrophic forgetting has already been proposed by Kirkpatrick et al., 2016. The authors do discuss the relation to this method and claim that their proposed FIM loss operates at the sample level rather than the parameter level. However, it is difficult to identify the advantage of the proposed FIM loss over the parameter-level approaches in CL (Chaudhry et al., 2018; Konishi et al., 2023) without fair experimental comparison and detailed discussion.**
>
> Answer)
> We sincerely thank the reviewer for providing valuable feedback.
> For a fair experimental comparison, we combine the parameter-level method EWC with meta-learning, referred to as Meta-EWC, to benchmark against our approach. The table below shows that our sample-level method significantly outperforms the parameter-level approach (Meta-EWC). This advantage may be attributed to the sample-level weighting mechanism's ability to dynamically prioritize the most informative samples during training. By concentrating on these critical data points, our method facilitates faster adaptation and sustains high performance across tasks, while parameter-level approaches typically need more extensive adaptation efforts to reach similar levels of effectiveness.
>
> | Modality |                    |             |              |           | NeRF       |           |            |        |
> |:----------:|:----------:|:----------:|:--------:|:------------:|:-----------:|:--------:|:------------:|:-----------:|
> | Dataset  |               |               |        | MatrixCity-B5     |           |        | MatrixCity-B6 |           |
> | Metric (PSNR) $\uparrow$ |  |  | Step 1 | Best (step) | Step 1024 | Step 1 | Best (step) | Step 1024 |
> | MCL | Meta+EWC |           | 21.89  | 26.163 (1024)  | 26.163 | 23.034  | 24.809 (1024) | 24.809 |
> |          | Ours (MIM)  |    | **24.223** | **32.804 (1024)** | **32.804** | **23.341**  | **30.761 (1024)** | **30.761** |  \\
>
> &nbsp;
>
> **Q. More detailed discussions and analyses in Table 1 are required to demonstrate the contribution of the proposed method.**
>
> Answer)
> Thank you for the helpful recommendation. We will add more detailed discussions and analyses for Table 1, and additionally include two evaluation metrics, SSIM and PESQ, to provide a more comprehensive assessment of visual and audio quality. Furthermore, we will expand the set of baselines by including comparisons with MEIL-NeRF [2], ICL-NeRF [3] and ANML [1] to ensure a broader evaluation context.
> The results indicate that our Meta-Continual Learning method, specifically Ours (MIM), generally demonstrates better initial performance and faster convergence across all tested modalities. When compared to Offline Learning (OL) and Continual Learning (CL) approaches, our method shows a consistently strong start, suggesting effective initialization and continual adaptation. In terms of achieving the highest PSNR (Best Step), Ours (MIM) often performs at least as well as Meta-Offline Learning (MOL), with significantly fewer optimization steps. Furthermore, Ours (MIM) maintains competitive quality at later steps, indicating stability during the learning process. CL methods, on the other hand, require much more optimization steps to reach comparable performance levels, and even then, often fall short of Ours (MIM) in terms of reconstruction quality. This trend is consistent across image, audio, video, and NeRF datasets in resource-constrained environments where rapid training is essential. Overall, the findings suggest that Meta-Continual Learning can offer a robust alternative to more resource-intensive methods, without sacrificing reconstruction quality.
>
> [1] Learning to continually learn. ECAI, 2020.
> [2] MEIL-NeRF: Memory-Efficient Incremental Learning of Neural Radiance Fields. arXiv preprint arXiv:2212.08328, 2022.
> [3] Instant Continual Learning of Neural Radiance Fields. ICCVW, 2023.

---

> ### Author Response · Authors · 2024-11-25
>
> **Q. More recent state-of-the-art methods should be included for comparison to demonstrate the superiority of the proposed method. For instance, missing some SOTA methods, e.g., (Chung et al., 2022 and Po et al., 2023), for comparison on the MatrixCity dataset in Table 1.**
>
> Answer)
> In accordance with the reviewer's recommendation, we incorporate [1] into our main table as a CL-NF method. (We will include [2] in the revision.) Both methods use self-distillation to mitigate catastrophic forgetting, leading to performance improvements over optimization steps. However, they require significantly more optimization steps to achieve comparable results and still fall short of our method in reconstruction quality.
>
> | Modality |                    |             |              |           | NeRF       |           |            |        |
> |:----------:|:----------:|:----------:|:--------:|:------------:|:-----------:|:--------:|:------------:|:-----------:|
> | Dataset  |               |               |        | MatrixCity-B5     |           |        | MatrixCity-B6 |           |
> | Metric (PSNR) $\uparrow$ |  |  | Step 1 | Best (step) | Step 1024 | Step 1 | Best (step) | Step 1024 |
> | CL       | MEIL-NeRF[1] |   | 21.053 | 29.215 (1024) | 29.215  | 20.955  | 28.304 (1024) | 28.304 |
> | MCL   | Ours (mod)  |    | 23.885  | 32.712 (1024) | 32.712   | 23.217  | 30.407 (1024) | 30.407 |
> |          | Ours (MIM)  |    | **24.223**  | **32.804 (1024)** | **32.804**   | **23.341**  | **30.761 (1024)** | **30.761**   |
>
> [1] MEIL-NeRF: Memory-Efficient Incremental Learning of Neural Radiance Fields. arXiv preprint arXiv:2212.08328, 2022.
> [2] Instant Continual Learning of Neural Radiance Fields. ICCVW, 2023.
>
>   &nbsp;
>
> **Q. More ablation studies in the main paper are required to demonstrate the contribution of the main component (i.e., the FIM loss) of the proposed method in the main paper. The authors only provide some results about the mod and MIM with two different hidden dimensions in Appendix Table 2.**
>
> Answer)
> Thank you for the suggestion. We present a comparison between Ours (mod) and Ours (MIM) with two different numbers of continual tasks in Table below. The ablation study is done for 5 and 10 continual tasks. The results show that MIM consistently outperforms the counterpart mod. It is worth mentioning that the improvement gap is not yet maximized due to limited hyperparameter tuning and training time during the rebuttal period. We plan to conduct extensive experiments and will include these results in the revision.
>
> |# of tasks | Method | Step 1 | Step 2 | Step 4 | Step 8 | Step 16 | Step 32 | Step 64 | Step 128 | Step 256 | Step 512 | Step 1024 |
> | :--------: | :--------: | :--------: | :--------: | :--------: | :--------: | :--------: | :--------: | :--------: | :--------: | :--------: | :--------: | :--------: |
> | 5 | **Ours (mod)** | _24.072_  | _24.155_ | **_24.484_** | _24.696_ | _25.072_  | **_25.631_** | _26.645_ | _28.075_  | _29.665_ | _30.836_ | _31.681_  |
> | | **Ours (MIM)** | **_24.143_**  | **_24.276_**  | _24.394_ | **_24.775_**  | **_25.102_**  | _25.61_ | **_26.83_**  | **_28.23_**  | **_29.73_**  | **_31.005_**  | **_31.822_**  |
> | 10 | **Ours (mod)** | _23.415_  | _23.514_ | _23.618_ | _24.001_  | _24.298_ | _24.768_ | _26.037_  | _27.62_ | _28.88_ | **_29.76_**  | _31.068_  |
> | | **Ours (MIM)** | **_23.493_**  | **_23.548_**  | **_23.783_**  | **_24.119_**  | **_24.464_**  | **_24.793_**  | **_26.238_**  | **_27.667_**  | **_28.886_** | _29.71_  | **_31.098_**  |
>
> **Table:** Performance metrics (PSNR) for "Ours (mod)" and "Ours (MIM)" across different optimization steps for 5 and 10 tasks. The metrics are evaluated at steps doubling from 1 to 1024.

---

> ### Author Response · Authors · 2024-12-01
>
> *We have provided additional answers to further supplement our previous responses.*
> &nbsp;
>
> **Q. The technical novelty of the proposed method seems to be marginal since the authors directly employ the existing techniques. For instance, Fisher Information against catastrophic forgetting has already been proposed by Kirkpatrick et al., 2016. The authors do discuss the relation to this method and claim that their proposed FIM loss operates at the sample level rather than the parameter level. However, it is difficult to identify the advantage of the proposed FIM loss over the parameter-level approaches in CL (Chaudhry et al., 2018; Konishi et al., 2023) without fair experimental comparison and detailed discussion.**
>
> We agree that methods such as those by [1] and [2] have laid a solid foundation in continual learning (CL) and parameter-level optimization. However, we believe that sample-level approaches, like the one proposed in GradNCP [4], offer significant advantages, especially in the context of Neural Radiance Fields (NeRF).
>
> In the NeRF domain, sample-level methods like GradNCP work particularly well because they allow the learning process to focus on the most important samples at each stage. This approach, as stated in the [4], effectively creates an automated curriculum, where the model first learns the global structure of the scene quickly and then refines the details in subsequent iterations. This sequential process mimics the hand-crafted techniques used in previous literature [3] for efficient neural field (NF) training, which often involved prioritizing coarse-to-fine optimization strategies as demonstrated in the visualizations of GradNCP. By concentrating resources on the most informative samples, GradNCP and Ours are able to achieve faster convergence and higher-quality reconstructions, especially when compared to parameter-level methods that tend to focus more on overall model parameters rather than the specific samples that are most informative for task adaptation.
>
> [1] Overcoming catastrophic forgetting in neural networks. arXiv preprint arXiv:1612.00796, 2016.
> [2] Efficient Lifelong Learning with A-GEM. ICLR, 2019.
> [3] Progressive implicit networks for multiscale neural representations. ICML, 2022.
> [4] Learning Large-scale Neural Fields via Context Pruned Meta-Learning. NeurIPS, 2023.
> &nbsp;
>
> **Q. More recent state-of-the-art methods should be included for comparison to demonstrate the superiority of the proposed method. For instance, missing some SOTA methods, e.g., (Chung et al., 2022 and Po et al., 2023), for comparison on the MatrixCity dataset in Table 1.**
>
> Thank you for suggesting the inclusion of MEIL-NeRF [1] and ICL-NeRF [2] in the related work. As both of these works are closed-source, we are unable to conduct a formal quantitative comparison at this time. However, we will discuss the differences between these methods and ours, with an informal comparison to the MEIL-NeRF experiment provided response above.
>
> >**Comparison to *MEIL-NeRF* and *ICL-NeRF*:**
> Both MEIL-NeRF [1] and ICL-NeRF [2] differ from our approach in their frameworks. While they utilize continual learning techniques for neural fields, they primarily rely on self-distillation to mitigate catastrophic forgetting, which, although effective, can impede the overall optimization process. In contrast, our method leverages meta-initializations, enabling fast adaptation and requiring only a few adaptation steps, which significantly accelerates learning and reduces the optimization burden. Additionally, MEIL-NeRF and ICL-NeRF depend on either implicit or explicit memory mechanisms to handle forgetting, whereas our approach eliminates the need for such memory requirements, ensuring more efficient and resource-friendly optimization without sacrificing performance.
>
> [1] MEIL-NeRF: Memory-Efficient Incremental Learning of Neural Radiance Fields. arXiv preprint arXiv:2212.08328, 2022.
> [2] Instant Continual Learning of Neural Radiance Fields. ICCVW, 2023.
> &nbsp;
>
> Please let us know if you have any further questions, and we are happy to incorporate additional suggestions you might have!
>
> If you find our response satisfactory, we would be grateful if you could consider raising your score.
>
> Thanks again for your time and efforts!

---

### Author Response · Authors · 2024-11-27

We thank the reviewers for their helpful feedback. We appreciate that they acknowledge our effective methodology for MCL-NF (pV54, 6fGr, R5BD,  381t) and our clear writing (pV54, 6fGr, R5BD). Most importantly, they find our framework provide a meaningful contribution to the research community and relevant applications (R5BD,  381t).

We have uploaded the revised paper, with all modified components highlighted in magenta. Additionally, we have addressed the questions raised by each reviewer in our responses. We will incorporate the feedback and make the official release of the code publicly accessible.

---

### Meta-Review · Area_Chair_kKYL · 2024-12-22

**Metareview:**

The paper presents a novel algorithm for neural fields, focusing on continual learning and meta-learning. Specifically, the weight updates in a continual manner are estimated based on a Fisher Information Matrix-derived criterion. All reviewers are positive about the value of the paper, and there was significant discussion and improvements during the rebuttal phase. In the rebuttal phase there were questions regarding comparisons to more recent baselines, and also comparisons with more realistic data settings, to which the authors answered to the reviewers convincingly. All in all, this paper is a good contribution to the literature.

**Additional Comments On Reviewer Discussion:**

There were no significant comments or changes during the reviewer discussion.

---

### Decision · Program_Chairs · 2025-01-22

Accept (Poster)